# Custom Gradient Estimators are Straight-Through Estimators in Disguise

## Abstract

Quantization-aware training comes with a fundamental challenge: the derivative of quantization functions such as rounding are zero almost everywhere and nonexistent elsewhere. Various differentiable approximations of quantization functions have been proposed to address this issue. In this paper, we prove that a large class of weight gradient estimators is approximately equivalent with the straight through estimator (STE). Specifically, after swapping in the STE and adjusting both the weight initialization and the learning rate in SGD, the model will train in almost exactly the same way as it did with the original gradient estimator. Moreover, we show that for adaptive learning rate algorithms like Adam, the same result can be seen without any modifications to the weight initialization and learning rate. These results reduce the burden of hyperparameter tuning for practitioners of QAT, as they can now confidently choose the STE for gradient estimation and ignore more complex gradient estimators. We experimentally show that these results hold for both a small convolutional model trained on the MNIST dataset and for a ResNet50 model trained on ImageNet.

## 1 Introduction

**The importance of quantized deep learning.** Quantized deep learning has gained significant attention as a means to address the demand for efficient deployment of deep neural networks on resource-constrained devices. Traditional deep learning models typically employ high-precision representations, consuming substantial computational resources and memory. Quantized deep learning techniques offer a compelling solution by reducing the precision of network parameters and activations. Although the Post-Training Quantization technique is easier to use to quantize any given model, Quantization-Aware Training (QAT) has been shown to provide higher quality results since quantized weights are updated throughout the training process (Nagel et al., 2021).

**Gradient estimators are needed in QAT.** QAT encounters a problem where the derivatives of quantization functions are zero or nonexistent everywhere. To sidestep this problem, practitioners use approximations of the quantization functions (known as *gradient estimators*) for backpropagation. The straight-through estimator is a common choice for this, but many believe it is better for a gradient estimator to more closely approximate the rounding function. We show that this belief is misguided.

**Our main contributions are as follows:**

1. A proof under minimal assumptions that all nonzero weight gradient estimators lead to approximately equivalent weight movement for non-adaptive learning rate optimizers (SGD, SGD + Momentum, etc.) when the learning rate is sufficiently small, after a change to weight initialization and learning rates has been applied.

2. A proof that for adaptive learning rate optimizers (Adam, RMSProp, etc.) the same result holds without any need for adjustment to the learning rate and weight initialization.

3. Empirical evidence demonstrating this result on both a small deep neural networked train on MNIST and a larger ResNet50 model trained on ImageNet.

**Value for practitioners:** Our findings reduce the burden of hyperparameter tuning for QAT. When learning rates are low, practitioners can now confidently choose the Straight Through Estimator

(Bengio et al., 2013) for weight gradient estimation and allocate their attention on problems like choosing the weight initialization scheme, learning rate, and optimization method.

## 2 BACKGROUND AND RELATED WORK

**The standard quantizer function.** The core operation in QAT is the application of a quantizer function to weights and activations, which transforms continuous, high-precision values into discrete, lower-precision representations. Quantization functions act elementwise on weight tensors $\mathbf{w}$, and can therefore be described by scalar functions on weights $w$. While there are many options for the arrangement of quantized values (Dettmers et al., 2023; Jung et al., 2019; Przewlocka-Rus et al., 2022; Oh et al., 2021; Liu et al., 2022), we will be focused on the most popular formulation, uniform quantization functions, which are defined by

$$Q(x) := \Delta \cdot \text{round}\left(\text{clip}\left(\frac{x}{\Delta}, l, u\right)\right) \qquad \text{where} \qquad \text{clip}(x, l, u) = \begin{cases} l & \text{if } x < l, \\ x & \text{if } l \leq x \leq u, \\ u & \text{if } x > u. \end{cases} \quad (1)$$

The problem of choosing $\Delta$, $l$, and $u$ is well-researched, and we cover common approaches in Appendix A.

**Boundary points.** We will refer to the sets of quantizer input values that map to a single output value as *quantization bins*. The boundaries of these bins are known as *boundary points*. We will use $w_+$ and $w_-$ to refer to the lower and upper boundary points for the bin containing weight $w$. One of these points must exist for each $w$, but outside of the representable range (see Appendix A) of the quantizer only one of the two will exist. Note that $w_+ - w_- = \Delta$ for all weights in the representable range.

**The Straight Through Estimator.** Because $Q'(x) = dQ/dx$ is zero almost everywhere and nonexistent elsewhere, vanilla gradient descent would never update the weights of a quantized model. The standard approach for addressing this issue is to approximate $Q(x)$ by a differentiable surrogate function $\hat{Q}$ and use its gradient $\hat{Q}'(x)$ for backpropagation. The derivative $\hat{Q}'$ is known as a *gradient estimator* (or *gradient approximation*). The earliest popular choice of gradient estimator is known as the *straight-through estimator* (Hinton, 2012; Bengio et al., 2013) or STE, defined by $\hat{Q}(x) = x$, $\hat{Q}'(x) = 1$. A strong theoretical justification for use of the STE is given in Yin et al. (2019).

**Piecewise linear estimators.** Piecewise linear (PWL) estimators have derivative $I_{[w_{min}, w_{max}]}$, where $I$ is the indicator function. They make $\hat{Q}$ more closely resemble $Q$ (Rastegari et al., 2016; Hubara et al., 2016; Zhang et al., 2022). The simplest way to define a PWL estimator for a multi-bit quantizer is to simply use Equation 1 with the round step removed, and in this case $[w_{min}, w_{max}]$ is exactly the representable range. This way, the behavior of PWL estimators more closely relate to the quantization function. In general, we will use $PWL_{w_{min}, w_{max}}(x) = \text{clip}(x, w_{min}, w_{max})$ to denote a PWL gradient estimator.

**STE and PWL lead to "gradient error".** The simple STE and PWL gradient estimators described above still leave a significant gap between the behavior of the forward pass and the surrogate forward pass. For this reason, researchers have proposed a large number of custom gradient estimators, often citing a high "gradient error" in the simpler choices of gradient estimators as motivation for their work. Gradient error is often described as the difference between $Q$ and $\hat{Q}$.

**An abundance of custom gradient estimators.** In order to solve the perceived problem of gradient error, many researchers have proposed gradient estimators that carry more complexity than the STE or PWL estimators. In Appendix B, we cite and describe **17** examples of custom gradient estimators in the quantization literature. Plots of some prominent examples are given in Figure 1.

## 3 GRADIENT DESCENT TERMINOLOGY FOR QAT

For a quantized model with gradient estimator $\hat{Q}$, the gradient value at step $t$ is $\nabla f(Q(w^{(t)}))\hat{Q}'(w^{(t)})$, where $f$ is the loss function of the model. Of course $f$ depends on the dataset and all other network weights, but we suppress this for notational convenience. Going forward, we will abbreviate

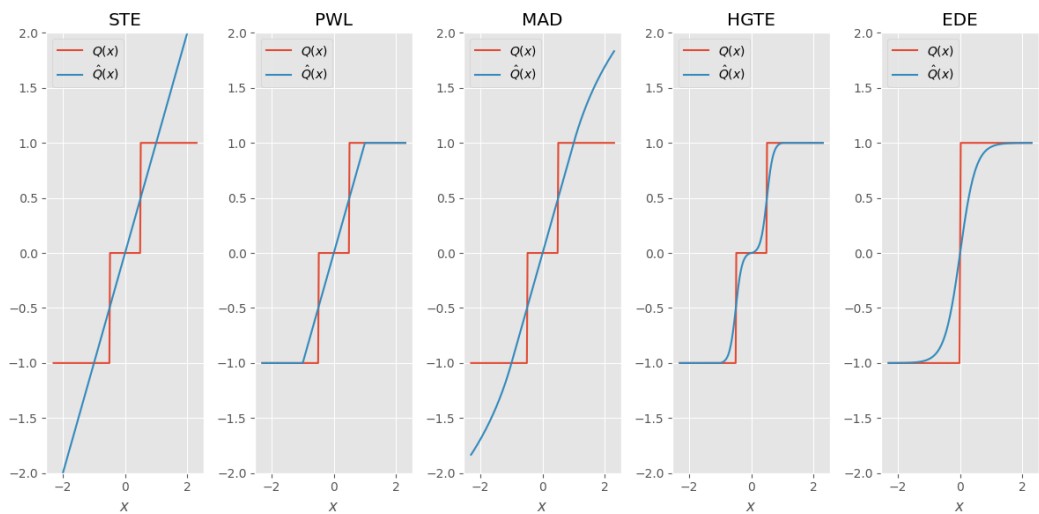

Figure 1: Gradient Estimators from left to right: STE (Hinton, 2012), PWL (Hubara et al., 2016), MAD (Sakr et al., 2022), HTGE (Pei et al., 2023), EDE (Qin et al., 2020). The EDE is for binary quantization, and the others are for multi-bit quantization.

$\nabla f(Q(w^{(t)}))$ as $\nabla f^{(t)}$. The weight update is expressed as

$$w^{(t+1)} = w^{(t)} + g^{(t)}(\nabla f^{(0)} \hat{Q}'(w^{(0)}), \ldots, \nabla f^{(t)} \hat{Q}'(w^{(t)}), \eta). \tag{2}$$

where $\eta$ is the learning rate. The notation for $g^{(t)}$ is borrowed from Andrychowicz et al. (2016). By defining $g^{(t)}$, we can recover all of the standard gradient descent algorithms, i.e. SGD, Adam, RMSProp, etc. In the simplest case, we have $g^{(t)}(\nabla f^{(t)} \hat{Q}'(w^{(t)}), \eta) = -\eta \nabla f^{(t)} \hat{Q}'(w^{(t)})$, which gives us the common SGD learning rule

$$w^{(t+1)} = w^{(t)} - \eta \nabla f^{(t)} \hat{Q}'(w^{(t)}). \tag{3}$$

The definition of $g^{(t)}$ for SGD with momentum is given in Appendix D. A more complex but highly popular learning rule is the Adam (Kingma and Ba, 2014) optimizer, which is defined with the above notation in Appendix E.

**Adaptive and non-adaptive algorithms.** Adam is an example of an *adaptive learning rate algorithm*, since the weight update steps are normalized by a computation on past gradient values. Other examples of adaptive learning rate methods are RMSprop (Hinton, 2012), Adadelta (Zeiler, 2012), AdaMax (Kingma and Ba, 2014), and AdamW (Loshchilov and Hutter, 2017), We refer to all other update rules, such SGD and SGD with momentum (Qian, 1999), as *non-adaptive learning rate algorithms*.

## 4    INTUITION

To aid the reader in developing intuition about our main results, we tell a brief story below.

**The Mirror Room story.** Imagine you are in a room with a glass wall. On the other side of the glass wall, there is a person in another room, larger than yours. You are standing at different positions in your respective rooms. Any time you take a step, this other person takes a step in the same direction, albeit with a different step length. You continue to move around, and you are rarely exactly across from this person, but any time you try to leave, this person leaves the room on the same side at the same time.

You realize that the glass wall is not a wall, it's a funhouse mirror. The person on the other side is you, but the picture is "warped" by the mirror.

**The Mirror Room is the quantization bin for two equivalent models.** The scenario described above is similar to the relationship between the motion of weights in a model ($\hat{Q}$-net) that uses a

complex gradient estimator $\hat{Q}$ and another ($STE$-net) that uses the $STE$ with the proper reconfigurations to match $\hat{Q}$-net. In the analogy, you are a weight in $STE$-net, your reflection is the weight in $\hat{Q}$-net. The room is a quantization bin, and the doors are the boundary points. The simultaneous exit of you and your reflection from the room parallels the synchronized quantized weights in both models, leading to identical gradients and training outcomes.

**The "Funhouse Mirror" effect of $M$ and $\hat{Q}$.**
In Section 5, we define a map $M$ that acts as a "funhouse mirror" mapping the weights of $\hat{Q}$-net to those of $STE$-net. Any initial weight $w^{(0)}$ in $\hat{Q}$-net is re-initialized to $M(w^{(0)})$ in $STE$-net, and the relationship $M(w_{\hat{Q}}) = w_{STE}$ approximately holds throughout training, where $w_{\hat{Q}}$ is a weight in $\hat{Q}$-net, and $w_{STE}$ is the corresponding weight in $STE$-net. Thus after the $\hat{Q}$-net weight takes a step, the $STE$-net weight moves in near lockstep after passing through the "funhouse mirror" of $M$. The fidelity of this approximation is given by $E^{(t)}$ (defined in Equation 5) at each step, which we show is small whenever the learning rate is small. Furthermore, since $M(w) = w$ whenever $w$ is a boundary point, these two weights will cross the quantization boudaries at nearly the same time. The bisimulation of the two models is justified by this property.

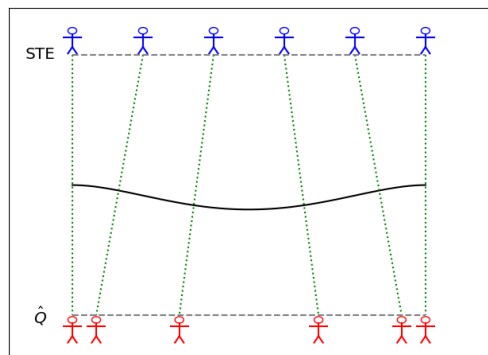

Figure 2: The funhouse mirror. The blue figure represents you (a weight in $STE$-net), and the red figure represents your reflection (a weight in $\hat{Q}$-net) on the other side. The reflections line up at the edge of the room.

# 5 MAIN RESULTS

In this section we formalize the realizations of Section 4 and provide our main mathematical results (1 and 2). Furthermore, this will show that much of the concern about "gradient error" is unfounded. We provide Theorem statements for both the SGD update rule and the Adam update rule, with proofs and generalizations in the Appendices. Note that all of the below results apply to weight quantizers. We do not address activation quantizers in this work.

## 5.1 DEFINITIONS AND NOTATION

**Cyclical gradient estimators.** We say that a gradient estimator $\hat{Q}$ for a uniform quantizer $Q$ is *cyclical* if $\hat{Q}$ is identical on each finite-length quantization bin, i.e. $\hat{Q}'(w) = \hat{Q}'(w + \Delta)$ whenever $w$ and $w + \Delta$ are inside a finite-length quantization bin (i.e. within the representable range). Most multi-bit gradient estimators proposed in the literature are cyclical. Binary gradient estimators are cyclical by default, since they have no finite quantization bins. Unless otherwise specified, we will assume that all gradient estimators are cyclical.

**Definitions of $\alpha$ and $M$.** We give two more definitions before presenting the details of the models we are comparing. These objects ($\alpha$ and $M$) will allow us to succinctly express the learning rate update and weight initialization update needed to mimic the behavior of a positive gradient estimator $\hat{Q}$ using only the STE. If $Q$ is a uniform multi-bit quantizer and $\hat{Q}$ is cyclical, we define the learning rate adjustment factor $\alpha$ and weight readjustment map $M$:

$$\alpha := \frac{\Delta}{\int_{w_-}^{w_+} \frac{ds}{\hat{Q}'(s)}} \qquad M(w) := w_b + \alpha \int_{w_b}^{w} \frac{ds}{\hat{Q}'(s)} \qquad (4)$$

Here $w_+$ and $w_-$ are adjacent boundary points, and $w_b$ is any standalone boundary point. Since $Q$ is uniform and $\hat{Q}$ is cyclical, the definition of $\alpha$ is independent of the choice of boundary points. If $Q$ is a binary quantizer, then $Q$ has only one boundary point, and we define $\alpha := 1$. Note that $\alpha$ is defined entirely by $\hat{Q}$, and can be computed at the outset of training. It may vary per-layer if the parameters

of $\hat{Q}$ do so. Intuitively it can be thought of as the ratio between the quantization bin size ($\Delta$) and the "effective bin size" of a gradient estimator $\hat{Q}$ (the denominator of Equation 4). The definition of $M$ is independent of the choice of $w_b$. We can think of $M$ as a function that maps a weight $w$ to a new point $M(w)$ whose relative distance from its left and right boundaries matches the relative "effective distance" (under $\hat{Q}$) between the boundary points and the original weight $w$.

**Definition of $\hat{Q}$-net and $STE$-net.** For both optimization techniques we consider (SGD and Adam) we will study two models, $\hat{Q}$-net and $STE$-net. The models can have any architecture, as long as they are equivalent. We will focus on corresponding weights $w_{\hat{Q}}^{(t)}$ and $w_{STE}^{(t)}$, respectively, at iteration $t$. We will denote the gradients of the loss function $f$ with respect to $Q(w_{\hat{Q}}^{(t)})$ and $Q(w_{STE}^{(t)})$ as $\nabla f_{\hat{Q}}^{(t)}$ and $\nabla f_{STE}^{(t)}$, respectively. The differences in gradient estimators, learning rates and weight initialization for both SGD and Adam are given in Tables 1 and 2, respectively.

<table>
<tr><td colspan="3">Table 1: $\hat{Q}$ and $STE$ Models for SGD</td><td colspan="3">Table 2: $\hat{Q}$ and $STE$ Models for Adam</td></tr>
<tr><td>**Model**</td><td>$\hat{Q}$**-net**</td><td>$STE$**-net**</td><td>**Model**</td><td>$\hat{Q}$**-net**</td><td>$STE$**-net**</td></tr>
<tr><td>Gradient Estimators</td><td>$\hat{Q}$</td><td>$STE$</td><td>Gradient Estimators</td><td>$\hat{Q}$</td><td>$STE$</td></tr>
<tr><td>Learning Rates</td><td>$\eta$</td><td>$\alpha\eta$</td><td>Learning Rates</td><td>$\eta$</td><td>$\eta$</td></tr>
<tr><td>Initial Weights</td><td>$w_{\hat{Q}}^{(0)}$</td><td>$M(w_{\hat{Q}}^{(0)})$</td><td>Initial Weights</td><td>$w_{\hat{Q}}^{(0)}$</td><td>$w_{\hat{Q}}^{(0)}$</td></tr>
</table>

**Comparison Metric.** We can quantify how the weights between $\hat{Q}$-net and $STE$-net differ using weight alignment error, which is defined as

$$E^{(t)} := \left| M\left(w_{\hat{Q}}^{(t)}\right) - w_{STE}^{(t)}\right| \quad \text{for SGD, and} \quad E^{(t)} := \left| w_{\hat{Q}}^{(t)} - w_{STE}^{(t)}\right| \quad \text{for Adam.} \quad (5)$$

$E^{(t)}$ measures how far off the weights are between the two models at iteration $t$, and starts at $E^{(0)} = 0$ due to our choice of initial weights in Tables 1 and 2. Furthermore, since $M$ preserves quantization bins, we have that $Q(w_{\hat{Q}}^{(t)}) = Q(w_{STE}^{(t)})$ whenever $E^{(t)}$ is small.

### 5.2 THEOREM STATEMENTS

Theorem 5.1 rigorously states contribution 1 for the SGD update rule (Equation 3). It states that after adjusting the learning rate of a model by $\alpha$ and re-initializing the weights by applying $M(w)$, a positive gradient estimator $\hat{Q}$ can be replaced by the STE with minimal differences in training.

**Theorem 5.1.** *Suppose that $E^{(t)}$ is the alignment error for $\hat{Q}$-net and $STE$-net with SGD (Table 1). Assume that the following hold:*

*5.1.1 $0 < L_- \le \hat{Q}'(w) \le L_+$ for all $w$. (**Bounded, positive gradient estimator**)*

*5.1.2 $\hat{Q}'(w)$ is $L'$-Lipschitz. (**Well-behaved gradient estimator**)*

*Then we have*

$$E^{(t+1)} \le E^{(t)} + \underbrace{\eta\alpha\left|\nabla f_{\hat{Q}}^{(t)} - \nabla f_{STE}^{(t)}\right|}_{\text{gradient error}} + \underbrace{\frac{L'}{2}\cdot\left(\frac{\eta L_+ \nabla f_{\hat{Q}}^{(t)}}{L_-}\right)^2}_{\text{convexity error}} \quad (6)$$

See Appendix C for a rigorous proof. The theorem only considers the standard gradient descent process. For a similar statement for a more general class of non-adaptive learning rate optimizers, see Appendix C. See Appendix D for a more specific result for SGD with momentum.

Theorem 5.2 rigorously proves contribution 2 for the Adam update rule (Equations 57-61). The result here is stronger than Theorem 5.1. When using the Adam update rule, the gradient estimator $\hat{Q}$ can be replaced by the STE *without* any update to the learning rate or weight initialization.

**Theorem 5.2.** *Suppose that $E^{(t)}$ is the alignment error for $\hat{Q}$-net and $STE$-net with Adam (Table 2). Assume that the following hold:*

  5.2.1  $0 < L_- \leq \hat{Q}'(w)$ *for all $w$. (**Lower bounded positive gradient estimator**)*

  5.2.2  $\hat{Q}'(w)$ *is $L'$-Lipschitz. (**Well-behaved gradient estimator**)*

*Then we have*

$$E^{(t+1)} \leq E^{(t)} + \underbrace{\left| g^{(t)}(\nabla f_{\hat{Q}}^{(0)}, \ldots, \nabla f_{\hat{Q}}^{(t)}, \eta) - g^{(t)}(\nabla f_{STE}^{(0)}, \ldots, \nabla f_{STE}^{(t)}, \eta) \right|}_{gradient\ error} + \underbrace{O(\eta^2)}_{convexity\ error} \ , \quad (7)$$

*where $g^{(t)}$ is the gradient update rule for Adam (see Equation 2 and Equations 57-61).*

See Appendix E for a rigorous proof. In Theorem 5.2, the exact definition of the $O(\eta^2)$ term is omitted due to its complexity. For a similar statement for a more general class of non-adaptive learning rate optimizers (not just the Adam optimizer), see Appendix E. For a discussion of Theorems 5.1 and 5.2 for learning rate schedules, see Appendix F.

### 5.3  ON THE ASSUMPTIONS AND IMPLICATIONS OF THEOREMS 5.1 AND 5.2

Theorems 5.1 and 5.2 rely on specific assumptions about the gradient estimator $\hat{Q}$. In this section, we break down these assumptions clearly. Furthermore, we describe how these theorems imply contributions 1 and 2.

**The assumptions are reasonable:** The upper bound on $\hat{Q}'$ in Assumption 5.1.1 is very mild. Gradient estimators with an unbounded derivative would likely cause training instability, and are not used in practice. Similarly, the authors are not aware of a gradient estimator that breaks Assumptions 5.1.2 and 5.2.2. In addition, the constants $L_-$, $L_+$, and $L'$ are usually quite small in practice (see Appendix H for calculations). The lower bound on $\hat{Q}'$ in Assumptions 5.1.1 and 5.2.1, however, is often broken in practice. In Appendix G, we describe how the Theorems still support contributions 1 and 2 in these cases.

**The bounds in Equations 6 and 7 are small:** In order to see how Theorems 5.1 and 5.2 provide contributions 1 and 2, we can closely examine each term in Equations 6 and 7. The gradient and convexity error in each equation together give a worst-case increase to $E^{(t)}$ at each training step. That is, as long as these terms are small, $\hat{Q}$-net and $STE$-net will train in a very similar manner. The convexity error terms are unavoidable errors, and are extremely small ($O(\eta^2)$) in practice. The gradient error terms, however, are $O(\eta)$, so they can be large if the gradients of the two models are misaligned. *However, since the gradient terms $\nabla f_{\hat{Q}}^{(t)}$ and $\nabla f_{STE}^{(t)}$ only depend on quantized weights, these terms will be zero at the beginning of training and remain small as long as $E^{(t)}$ remains small.*

**The claim is nontrivial:** Note that these theorems do *not* simply say that when the learning rate is small, the models change very little, and therefore $\hat{Q}$-net and $STE$-net are aligned. Since the irreducible error term is quadratic in $\eta$, the misalignment at each step is small *relative to the learning rate itself*.

**The claim applies to networks of any size:** The Theorems only give bounds for the error in a single network weight, but can be applied to each weight independently in a multi-weight network. Of course, the trajectories of weights in a neural network are not independent, but luckily in our case the weight trajectories only depend on the quantized versions of the other network weights. To see this, note that the only terms in Equations 6 and 7 that depend on other network weights are the gradient error terms. As stated earlier, these gradient terms only depend on quantized weights, so we do not need perfect alignment in other latent network weights in order to keep the error terms in these Equations small. Since the gradient error terms can depend on all other quantized weights in the network, larger models are at a greater risk of weight misalignment. However, this is more a property

of large models than of gradient estimators: any two large models that have only a small difference in hyperparameter configurations but otherwise equivalent training setups will have potentially large step-by-step divergences in weight alignment. And the fundamental difference in training induced by a gradient estimator is indeed small, since in Equations 6 and 7, the true source of all misalignment is an $O(\eta^2)$ term. This is supported by our experiments in Section 6.

**The claim applies to the entire trajectory.** While the Theorems give a bound for $E^{(t+1)}$ at each step in terms of $E^{(t)}$, we can apply out results to get absolute bounds on $E^{(t+1)}$ throughout the entire trajectory. For example, if the assumptions of Theorem 5.1 hold, then we can repeatedly apply Equation 6 for decreasing values of $t$ to obtain

$$E^{(t+1)} \leq \eta\alpha \sum_{i=1}^{t} \left| \nabla f_{\hat{Q}}^{(i)} - \nabla f_{STE}^{(i)} \right| + \frac{L'\eta^2 L_+^2}{2L_-^2} \cdot \sum_{i=1}^{t} \left( \nabla f_{\hat{Q}}^{(t-1)} \right)^2 .$$

A similar statement holds for Theorem 5.2. While this bound grows linearly with $t$ in the worst case, the same can be said about an $E^{(t)}$ metric comparing weights for two models that differ only by a very small factor in the learning rate. This motivates the "lr-tweak" experiments in Section 6.

### 5.4 THEOREM 5.1 PROOF SKETCH

The proof of Theorem 5.1 in its full generality requires heavy notation and somewhat obscures the simple point of the Theorem. Because of this, we provide a sketch of proof below.

*Sketch of Theorem 5.1 proof.* We have for all $t$,

$$E^{(t+1)} = \left| M\left( w_{\hat{Q}}^{(t+1)} \right) - w_{STE}^{(t+1)} \right| \tag{8}$$

$$\text{(Expand terms)} = \left| M\left( w_{\hat{Q}}^{(t)} - \eta\nabla f_{\hat{Q}}^{(t)} \hat{Q}'\left( w_{\hat{Q}}^{(t)} \right) \right) - \left( w_{STE}^{(t)} - \eta\alpha\nabla f_{STE}^{(t)} \right) \right| \tag{9}$$

$$\text{(Taylor's Thm.)} = \left| M\left( w_{\hat{Q}}^{(t)} \right) - \eta\nabla f_{\hat{Q}}^{(t)} \hat{Q}'\left( w_{\hat{Q}}^{(t)} \right) M'\left( w_{\hat{Q}}^{(t)} \right) - \left( w_{STE}^{(t)} - \eta\alpha\nabla f_{STE}^{(t)} \right) + O(\eta^2) \right| \tag{10}$$

$$\text{(Apply Eq. 13)} = \left| M\left( w_{\hat{Q}}^{(t)} \right) - \eta\alpha\nabla f_{\hat{Q}}^{(t)} - \left( w_{STE}^{(t)} - \eta\alpha\nabla f_{STE}^{(t)} \right) + O(\eta^2) \right| \tag{11}$$

$$\text{(Triangle Ineq.)} \leq E^{(t)} + \eta\alpha \left| \nabla f_{\hat{Q}}^{(t)} - f_{STE}^{(t)} \right| + O(\eta)^2 \tag{12}$$

Here Equation 10 follows from Taylor's Theorem. Equation 11 follows from Equation 13 below

$$\frac{\partial M}{\partial w}(w) = \alpha \cdot \frac{1}{\hat{Q}'(w)}, \tag{13}$$

and Equation 12 follows from the triangle inequality. The complete proof simply requires writing out an explicit form for the $O(\eta^2)$ term, and is given in detail in Appendix C. $\qquad\square$

## 6 EXPERIMENTAL RESULTS

Here we demonstrate our main results on practical models. The general strategy we will take is to implement $\hat{Q}$-net and $STE$-net for a specific model architecture and compare on a variety of metrics to demonstrate the following:

    A. $\hat{Q}$-net and $STE$-net train in almost exactly the same way.

    B. If we do not apply the weight re-initialization of Theorem 5.1, we do not see the same results.

### 6.1 MODELS AND TRAINING SETUP

**Models and Datasets**. We use two model architecture/dataset pairs:

1. A simple three-layer quantized convoluational archicture proposed in Chollet (2021) for image classification on the MNIST dataset, which gives a uniform weight distribution with the variance recommended in He et al. (2015) trained on a CPU.

2. ResNet50 (He et al., 2016) on the ILSVRC 2012 ImageNet dataset (Deng et al., 2009), which showcases generality to a more complex model and dataset trained on a TPU. We used a fully deterministic version of the Flax example library (Flax contributors, 2024).

**Weight Initialization and Quantizers:** We initialize the weights of $\hat{Q}$-net using He Uniform Initialization[1]. For quantization, we use a uniform weight quantizer with representable range limits given by bounds of the weight initialization distribution. We do not quantize activations. We focus primarily on two-bit weight quantization, and note that results are similar for 1-bit and 4-bit quantization. For gradient estimation, we use the $\hat{Q}$ given by the HTGE (Pei et al., 2023) gradient estimator formula with shape parameter $t$ set to 5.5 times the maximum value from the weight initialization distribution. This value was chosen so that $\hat{Q}$ differs significantly from the STE, but not so significantly that parts of $\hat{Q}$ become essentially flat.

**Optimization techniques.** For optimization techniques on both models, we consider both SGD with momentum$= 0.9$ and Adam with $\beta_1 = 0.9$ and $\beta_2 = 0.95$. For all experiments, we use a cosine decay learning rate schedule (Loshchilov and Hutter, 2016) with a linear learning rate warmup (Goyal et al., 2017) for 2% of training epochs. The reported learning rate for each model is the initial learning rate for the cosine decay. We use a learning rate of 0.001 for our default MNIST SGD with momentum model, and 0.0001 for our default MNIST Adam model. For the ResNet50 on ImageNet model we apply the standard learning rate schedule implemented in Flax contributors (2024) with a configured learning rate of 0.0001, for Adam and 0.001 for SGD and otherwise default parameters.

**Identical Initial Training period.** For the ImageNet-ResNet setup, we ensured that the first 10% of training for $\hat{Q}$-net and $STE$-net were identical. To do this, we trained $STE$-net by first training $\hat{Q}$-net for the first 10 of 100 epochs, and then applied $M$ to the weights and optimizer state and switched the model's quantizer for the STE before continuing training. This was applied for all model comparisons.

### 6.2 METRICS.

We use two metrics in order to establish Points A and B. Both of these compare $STE$-net weights to $\hat{Q}$-net weights. In addition to the metrics below, we also report accuracy and loss statistics for all models.

**Quantized Weight Agreement.** At the end of training the complete set of quantized weights is calculated for both models and compared. We report the proportion of quantized weights that are the same for both models.

**Normalized Weight Alignment Error ($\bar{\mathrm{E}}$).** For each pair of models, we compute the average value of $E^{(T)}$ for the final training step $T$ over all weights. Note that Equation 5 gives two definitions of $E$, and for each model pair we use the version that matches the weight initialization setup, which gives $E^{(0)} = 0$ for all model pairs. Each $E^{(T)}$ is normalized by the length of the representable range, so that a value of 100% indicates that the two models' weights are on opposite sides of the representable range. We denote the average as $\bar{E}$ for all model pairs.

### 6.3 RESULTS

**Tables for Points A and B:** We provide all metrics for both the default SGD and Adam models described in Section 6.1 within in Table 4, with detailed interpretations for the $\bar{E}$ metric in Table 3. Note that Adam does not have an "unadjusted" case, since there is no need for weight initialization adjustment when Adam is used.

**Point A is validated.** The standard comparison between $\hat{Q}$-net and $STE$-net is labeled as "baseline". We compute metrics between a $\hat{Q}$-net model and the same model with a learning rate increase of 1%

---

[1]`https://www.tensorflow.org/api_docs/python/tf/keras/initializers/HeNormal`

Table 3: Normalized weight alignment metric $\bar{E}$ for MNIST model with SGD + Momentum, including descriptions and interpretations for all four experiment types. This table serves as a guide for interpreting Table 4.

| Experiment Name | Experiment Description | $\bar{E}$ | Interpretation/Comparison to Baseline |
|---|---|---|---|
| baseline | $\hat{Q}$ vs. STE | 0.515% | Baseline |
| lr-tweak | $\hat{Q}$ vs. $\hat{Q}$ with 1% learning rate increase | 0.572% | Replacing $STE$-net with $\hat{Q}$-net is about as impactful as a small change to $\eta$ (A). |
| unadjusted | $\hat{Q}$ vs. STE *without* reinitializing weights | 2.52% | The two models only see the same weight movement if weights are re-initialized according to $M$ (B). |

Table 4: Alignment metrics for SGD (S) and Adam (A). Results for the MNIST model are shown on the left, and results for ResNet50 trained on ImageNet are shown on the right.

| Experiment Name | $\bar{E}$ | Quantized Weight Agreement | Experiment Name | $\bar{E}$ | Quantized Weight Agreement |
|---|---|---|---|---|---|
| baseline (S) | 0.515% | 98.31% | baseline (S) | 5.42% | 68.94% |
| lr-tweak (S) | 0.572% | 98.66% | lr-tweak (S) | 5.46% | 75.64% |
| unadjusted (S) | 2.52% | 96.53% | unadjusted (S) | 7.88% | 67.53% |
| baseline (A) | 2.81% | 94.42% | baseline (A) | 7.18% | 72.22% |
| lr-tweak (A) | 1.74% | 95.4% | lr-tweak (A) | 4.99% | 76.32% |

(chosen arbitrarily and only once), reported with the label "lr-tweak". This serves as an example of a "small change" to a model that the reader may be more familiar with, providing additional context about the scale of the metric results and supporting Point A. For both the MNIST and ImageNet models, the alignment between $\hat{Q}$-net and $STE$-net is similar to the alignment expected from a 1% learning rate change.

**Point B is validated.** We report alignment measurements between $\hat{Q}$-net and $STE$-net *without* the weight and learning rate adjustments described in Theorem 5.1 using the label "unadjusted". The alignment worsens for both the MNIST model and the ResNet model when removing the weight reinitialization by $M$.

**Weight Alignment.** For a visual of the weight alignment phenomenon, see Figure 3 in Appendix J.

**There is almost no difference in training accuracy.** Standard training metrics for both $\hat{Q}$-net and $STE$-net are given in Table 5 for both optimizers and both models we consider. This table shows that the two models have very similar train and test metrics, indicating that replacing $\hat{Q}$ with the STE is of minimal impact after applying the appropriate weight initialization and learning rate adjustments. As expected, the alignment is stronger for the smaller model.

## 7 IMPLICATIONS

Here we discuss the implications of this work on the existing literature and future practice and research.

**For practitioners.** The main message for practitioners is simple, and depends on the optimization strategy used as follows:

- **SGD and other non-adaptive optimizers:** In this case, if the learning rate is sufficiently small and you wish to tweak the gradient estimator, you can instead apply a corresponding weight re-initialization and learning rate adjustment to a model with the STE or PWL estimator and see nearly the same training procedure. The proof and related assumptions are given in Theorem C.1.

Table 5: Loss and Accuracy differences between $\hat{Q}$-net and $STE$-net with SGD (S) and Adam (A). Results for the MNIST model are shown on the left, and results for ResNet50 trained on ImageNet are shown on the right. For both SGD (S) and Adam (A) and both models, differences are small.

|  | Train acc | Train loss | Val acc | Val loss |  | Train acc | Train loss | Val acc | Val loss |
|---|---|---|---|---|---|---|---|---|---|
| STE (S) | 97.05% | 0.1439 | 97.08% | 0.1417 | STE (S) | 68.94% | 1.3370 | 69.83% | 1.2227 |
| $\hat{Q}$ (S) | 96.98% | 0.1483 | 97.14% | 0.1468 | $\hat{Q}$ (S) | 68.51% | 1.3365 | 68.77% | 1.2793 |
| **Diff** | **-0.06%** | **0.0044** | **0.06%** | **0.0051** | **Diff** | **0.43%** | **0.0005** | **-1.06%** | **-0.0566** |
| STE (A) | 97.56% | 0.1270 | 97.66% | 0.1257 | STE (A) | 69.78% | 1.2876 | 70.01% | 1.2209 |
| $\hat{Q}$ (A) | 97.63% | 0.1254 | 97.58% | 0.1245 | $\hat{Q}$ (A) | 69.02% | 1.3153 | 69.37% | 1.2490 |
| **Diff** | **0.07%** | **-0.0016** | **-0.08%** | **-0.0013** | **Diff** | **-0.77%** | **0.0277** | **-0.65%** | **0.0281** |

- **Adam and other adaptive optimizers:** In this case, when the learning rate is sufficiently small, the only gradient estimators you need consider are the STE and PWL estimators. The proof and related assumptions are given in Theorem E.1.

**For researchers.** For future research, we hope that this work will inspire further study on processes for updating quantized model parameters that are fundamentally different from the use of gradient estimators, and therefore immune to the arguments of this paper. This may include novel computations on gradients that diverge from the standard chain rule (Lee et al., 2021; Wangl et al., 2023), optimizers specially designed for QAT (Helwegen et al., 2019), or even methods that do not involve gradient computations at all (Takemoto et al., 2023). As for the existing literature, our message is that the concern about "gradient error" should not be considered in the future. For discussions about potential avenues for future work, see Appendix I.

## 8 DISCUSSION: WHY ARE SO MANY GRADIENT ESTIMATORS PUBLISHED?

A natural question that a reader may have concerning past research is this: If the choice of gradient estimator is so irrelevant, why is there so much research that proposes new gradient estimators and demonstrates improved performance with their aid? There are several potential answers to this.

**Performance differences are due to implicit weight initialization and learning rate differences.** The simplest explanation is that their gradient estimation techniques happen to have implictly uncovered a superior weight re-initialization and learning rate adjustment, as indicated by Theorem 5.1.

**Activation gradient estimators make a difference.** Another answer could be that the performance improvements were due to changes in quantized activation gradient estimators, which cannot be equated to the STE.

**Learning rates are too high to see the equivalence.** It is possible that the learning rates in these experiments were too high to see an equivalence between their gradient estimators and the STE. This is a limitation of our main argument, but we expect that this counter-argument will not stand the test of time, since by our main results, the higher learning rate masks the fact that models with novel $\hat{Q}$ and the STE are still approximating the same process.

**Gradient estimators are proposed alongside other innovations, making them hard to evaluate in isolation.** The most common situation is that novel gradient estimators $\hat{Q}$ are introduced simultaneously with further changes to the learning recipe. Some allow the parameters of $Q$ and $\hat{Q}$ to be learnable through gradient descent or explicit computations on the weights, or adjust them on a schedule (See Appendix A). Others, such as DSQ (Gong et al., 2019), use $\hat{Q}$ on the forward pass and gradually update $\hat{Q}$ to more closely approximate $Q$. Lin et al. (2020) contributes a process for rotating the entire weight vector to align with the binarized weight vector. Bi-Real Net Liu et al. (2018) also includes a trick with network activations to increase the representational capacity of the model. In addition to the Error Decay Estimator, Qin et al. (2020) describes a method for maximizing the entropy of quantized parameters to ensure higher parameter diversity.

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

## A    CHOOSING QUANTIZATION PARAMETERS

The clipping bounds $l$ and $u$ are determined by the number of bits $b$ in the quantized representation and the desired number of representable values in the positive and negative range of the quantizer. This range of weight values is referred to as the *representable range* (or *quantization range*) of the quantizer, and can be computed as $[\Delta \cdot l, \Delta \cdot u]$. Large $\Delta$ values allow for large $w$ values to avoid the clip step, whereas small values give small $w$ values a more granular representation. These parameters are either learned (Esser et al., 2019; Choi et al., 2018; Gong et al., 2019) or set by the user. For $b > 1$, $l$ and $u$ are often chosen as $l = -2^{b-1}$, $u = 2^{b-1} - 1$ for symmetric quantization and $l = 0$, $u = 2^b - 1$ for asymmetric quatization. $\Delta$ is often chosen uniformly per-channel or per-token, based off of latent weight data $W$. It is sometimes set as $\max(|W|)/(2^b - 1)$, or is chosen to minimize a loss function (such as MSE or cross entropy (Nagel et al., 2021)) comparing $W$ and $Q(W)$. For binary quantization $(b = 1)$, $Q(w)$ is typically a sign function (Nagel et al., 2021; Gholami et al., 2021; Rokh et al., 2022), and there is no representable range. For binary PWL estimators, a common choice is to use Equation 1 and simply set $\Delta = 1$ and $[w_{min}, w_{max}] = [-1, 1]$ (Sayed et al., 2023).

## B    DETAILED OVERVIEW OF CUSTOM GRADIENT ESTIMATORS

**Custom binary gradient estimators.** A substantial amount of research has gone into custom gradient estimators. Many choices (Sakr et al., 2018; Darabi et al., 2018; Liu et al., 2018; Xu and Cheung, 2019; Qin et al., 2020; Lin et al., 2020; Xu et al., 2021) for binary gradient estimators are described in (Yuan and Agaian, 2021). A popular estimator is the "Error Decay Estimator" (EDE) of (Qin et al., 2020), which uses an evolving $\tanh$ function to approximate the sign function.

**Custom gradient estimators.** The hyperbolic tangent gradient estimator (HTGE) (Pei et al., 2023) gives a piecewise function locally described by tanh functions. Its definition is given by the standard quantization function 1, where the round operation is replaced with

$$H(x) = \frac{a + b}{2} + \frac{\tanh\left(t\left(x - \frac{a+b}{2}\right)\right)}{2}$$

where $a = \text{floor}(x)$ and $b = \text{ceil}(x)$. Here $t$ is the sharpness parameter. This approximation is used for both the forward and backward pass of $Q$ in Differentiable Soft Quantization (DSQ) (Gong et al., 2019). Similar approaches to the HTGE use a sum of sigmoid functions (Yang et al., 2019) and a distance-weighted piecewise linear combination of the outputs of $Q$ (Kim et al., 2021) to approximate $Q$. These techniques make up the most common choices of gradient estimators, which justifies our choice of HTGE for our experiments. The gradient computation in Kim et al. (2020) leverages a special choice of $\hat{Q}$ based on the distance between the full-precision weight and its quantized version. Zhang et al. (2023) proposes a gradient estimator that includes an extra parameter that attempts to allow the quantization strategy to work well for both low-bit and high-bit quantization. Zhou et al. (2016) uses the STE for the round function, but replaces the clip function in the forward pass with a modified $\tanh$ function, which affects the gradient calculations as well. Sakr et al. (2022) introduces a choice for $\hat{Q}$ known as "Magnitude Aware Differentiation" (MAD) that matches the STE on the representable range of the quantizer and a reciprocal function outside of this range. More recently, Schaefer et al. (2024) proposed a gradient estimation method based on the inverse tanh function, and Yang et al. (2024) described a gradient estimator based on the cosine function. See Figure 1 for examples of several gradient estimators.

**Implications of our main results.** In light of our results 1 and 2, we can sometimes equate these addition algorithms with more well-known training strategies. For example, Qin et al. (2020) proposes a schedule for a $\tanh$-based gradient estimator to gradually approach a sign function throughout training. Since they use SGD in their experiments, we can think of each update to sharpen the gradient estimator as an effective "shifting" of the weights according to the function defined in Equation 4. This particular shift will push most weights away from 0, which has an effect similar to slowing down the learning rate. Thus this adaptive gradient estimation technique is similar to a standard learning rate decay schedule.

## C  PROOF OF THEOREM 5.1

Proving Theorem 5.1 will require several steps. First, in Theorem C.1 we prove a general statement that allows us to bound the increase in weight alignment error at each training step for any non-adaptive learning rate optimization strategy. This will allow us to quickly prove Theorem 5.1, and will also simplify the proof of a similar statement for SGD with momentum, which will be given in Appendix D.

Theorem C.1 applies to gradient update rules that satisfy a special property in Assumption C.1.3. We will show later in this section that this holds for the SGD formula defined in 3, and in Appendix D for SGD with momentum. Similar proofs show that it holds for a large class of non-adaptive learning rate gradient update rules.

**Theorem C.1.** *Suppose that*

$$E^{(t)} := \left| M\left(w_{\hat{Q}}^{(t)}\right) - w_{STE}^{(t)} \right| \tag{14}$$

*is the alignment error for $\hat{Q}$-net and $STE$-net with gradient estimators, learning rates, and initial weights given by Table 1. Suppose that Assumptions 5.1.1 and 5.1.2 hold and the model weights are updated according to Equation 2 for some function $g^{(t)}$. In addition, suppose that*

*C.1.3  For each $t$, the quantity*

$$\left| \frac{g^{(t)}(\nabla f_{\hat{Q}}^{(0)} \hat{Q}'(w^{(0)}), \dots, \nabla f_{\hat{Q}}^{(t)} \hat{Q}'(w^{(t)}), \eta)}{\hat{Q}'(w^{(t)})} - g^{(t)}(\nabla f_{\hat{Q}}^{(0)}, \dots, \nabla f_{\hat{Q}}^{(t)}, \eta) \right| = O(c(\eta)). \tag{15}$$

*Then we have*

$$E^{(t+1)} \leq E^{(t)} + \left| \alpha g^{(t)}(\alpha \nabla f_{\hat{Q}}^{(0)}, \dots, \nabla f_{\hat{Q}}^{(t)}, \eta) - g^{(t)}(\nabla f_{STE}^{(0)}, \dots, \nabla f_{STE}^{(t)}, \alpha \eta) \right| + \tag{16}$$

$$\frac{L'}{2} \cdot \left( \frac{g^{(t)}(\nabla f_{\hat{Q}}^{(0)} \hat{Q}'(w^{(0)}), \dots, \nabla f_{\hat{Q}}^{(t)} \hat{Q}'(w^{(t)}), \eta)}{L_-} \right)^2 + O(c(\eta)) \tag{17}$$

*Proof.* By Equation 2, we have

$$E^{(t+1)} = \left| M\left(w_{\hat{Q}}^{(t+1)}\right) - w_{STE}^{(t+1)} \right| \tag{18}$$

$$= \left| M\left(w_{\hat{Q}}^{(t)} + g^{(t)}(\nabla f_{\hat{Q}}^{(0)} \hat{Q}'(w^{(0)}), \dots, \nabla f_{\hat{Q}}^{(t)} \hat{Q}'(w^{(t)}), \eta)\right) - \tag{19}$$

$$\left( w_{STE}^{(t)} + g^{(t)}(\nabla f_{STE}^{(0)}, \dots, \nabla f_{STE}^{(t)}, \alpha \eta) \right) \right| \tag{20}$$

$$= \left| M\left(w_{\hat{Q}}^{(t)}\right) + g^{(t)}(\nabla f_{\hat{Q}}^{(0)} \hat{Q}'(w^{(0)}), \dots, \nabla f_{\hat{Q}}^{(t)} \hat{Q}'(w^{(t)}), \eta) M'\left(w_{\hat{Q}}^{(t)}\right) - \tag{21}$$

$$\left( w_{STE}^{(t)} + g^{(t)}(\nabla f_{STE}^{(0)}, \dots, \nabla f_{STE}^{(t)}, \alpha \eta) \right) + R \right| \tag{22}$$

$$= \left| M\left(w_{\hat{Q}}^{(t)}\right) + \alpha g^{(t)}(\nabla f_{\hat{Q}}^{(0)} \hat{Q}'(w^{(0)}), \dots, \nabla f_{\hat{Q}}^{(t)} \hat{Q}'(w^{(t)}), \eta) / \hat{Q}'\left(w_{\hat{Q}}^{(t)}\right) - \tag{23}$$

$$\left( w_{STE}^{(t)} + g^{(t)}(\nabla f_{STE}^{(0)}, \dots, \nabla f_{STE}^{(t)}, \alpha \eta) \right) + R \right| \tag{24}$$

$$= \left| M\left(w_{\hat{Q}}^{(t)}\right) + \alpha g^{(t)}(\nabla f_{\hat{Q}}^{(0)}, \dots, \nabla f_{\hat{Q}}^{(t)}, \eta) + O(c(\eta)) - \tag{25}$$

$$\left( w_{STE}^{(t)} + g^{(t)}(\nabla f_{STE}^{(0)}, \dots, \nabla f_{STE}^{(t)}, \alpha \eta) \right) + R \right| \tag{26}$$

$$\leq E^{(t)} + \left| \alpha g^{(t)}(\nabla f_{\hat{Q}}^{(0)}, \dots, \nabla f_{\hat{Q}}^{(t)}, \eta) - g^{(t)}(\nabla f_{STE}^{(0)}, \dots, \nabla f_{STE}^{(t)}, \alpha \eta) \right| + \tag{27}$$

$$|R| + O(c(\eta)) \tag{28}$$

Here Equation 22 follows from Taylor's Theorem, where $R$ is the remainder term. Equation 24 follows from Equation 13, and Equation 26 follows from Assumption C.1.3. Equation 28 follows from the triangle inequality. By Lemma 2.1 of Zhang (2023), we can bound $R$ by

$$|R| \le \frac{L'}{2L_-^2} \left( g^{(t)}(\nabla f_{\hat{Q}}^{(0)} \hat{Q}'(w^{(0)}), \dots, \nabla f_{\hat{Q}}^{(t)} \hat{Q}'(w^{(t)}), \eta) \right)^2, \tag{29}$$

To see this, we need to show that $M'$ is Lipschitz continuous with Lipschitz constant $L'/L_-^2$. This holds since for any $w, v \in \mathbb{R}$,

$$|M'(w) - M'(v)| = \left| \frac{1}{Q'(w)} - \frac{1}{Q'(w)} \right| = \left| \frac{Q'(v) - Q'(w)}{Q'(w)Q'(v)} \right| \le \frac{L'}{L_-^2} |w - v|.$$

In the last step we use both Assumptions 5.1.1 and 5.1.2. Putting this all together, we have Equation 17. $\qquad\square$

We can now apply Theorem C.1 for the SGD update rule (Equation 3) to give a proof of Theorem 5.1.

*Proof of Theorem 5.1.* To prove Theorem 5.1, we first show that Assumption C.1.3 holds for the SGD update rule with $c(\eta) = 0$. We have

$$\left| \frac{g^{(t)}(\nabla f_{\hat{Q}}^{(0)} \hat{Q}'(w^{(0)}), \dots, \nabla f_{\hat{Q}}^{(t)} \hat{Q}'(w^{(t)}), \eta)}{\hat{Q}'(w^{(t)})} - g^{(t)}(\nabla f_{\hat{Q}}^{(0)}, \dots, \nabla f_{\hat{Q}}^{(t)}, \eta) \right| = \tag{30}$$

$$\left| \frac{\eta \nabla f_{\hat{Q}}^{(t)} \hat{Q}'(w^{(t)})}{\hat{Q}'(w^{(t)})} - \eta \nabla f_{\hat{Q}}^{(t)} \right| = 0. \tag{31}$$

Now we can apply Theorem C.1. We have

$$E^{(t+1)} \le E^{(t)} + \left| \alpha g^{(t)}(\alpha \nabla f_{\hat{Q}}^{(0)}, \dots, \nabla f_{\hat{Q}}^{(t)}, \eta) - g^{(t)}(\nabla f_{STE}^{(0)}, \dots, \nabla f_{STE}^{(t)}, \alpha \eta) \right| + \tag{32}$$

$$\frac{L'}{2} \cdot \left( \frac{g^{(t)}(\nabla f_{\hat{Q}}^{(0)} \hat{Q}'(w^{(0)}), \dots, \nabla f_{\hat{Q}}^{(t)} \hat{Q}'(w^{(t)}), \eta)}{L_-} \right)^2 + O(c(\eta)) \tag{33}$$

$$= E^{(t)} + \eta \alpha \left| \nabla f_{\hat{Q}}^{(t)} - \nabla f_{STE}^{(t)} \right| + \frac{L'}{2} \cdot \left( \frac{\eta \nabla f_{\hat{Q}}^{(t)} \hat{Q}'(w^{(t)})}{L_-} \right)^2 + 0 \tag{34}$$

$$\le E^{(t)} + \eta \alpha \left| \nabla f_{\hat{Q}}^{(t)} - \nabla f_{STE}^{(t)} \right| + \frac{L'}{2} \cdot \left( \frac{\eta L_+ \nabla f_{\hat{Q}}^{(t)}}{L_-} \right)^2 \tag{35}$$

This gives us Equation 6, as desired. $\qquad\square$

## D   THEOREM 5.1 FOR SGD WITH MOMENTUM

Here we give a version of Theorem 5.1 for stochastic gradient descent with momentum. The weight update rule for this learning algorithm is given by

$$g^{(t)}(\nabla f^{(0)} \hat{Q}'(w^{(0)}), \dots, \nabla f^{(t)} \hat{Q}'(w^{(t)}), \eta) = -\eta m_t \tag{36}$$

where $m_t$ is defined recursively as

$$m_t = \beta m_{t-1} + (1 - \beta) \nabla f^{(t)} \hat{Q}'(w^{(t)}) \tag{37}$$

for a hyperparameter $\beta \in [0, 1)$, which is often set to 0.9 or a similar value (Ruder, 2016). We can expand this recursive definition, and obtain the single rule

$$g^{(t)}(\nabla f^{(0)} \hat{Q}'(w^{(0)}), \dots, \nabla f^{(t)} \hat{Q}'(w^{(t)}), \eta) = -\eta(1 - \beta) \sum_{i=0}^{t} \beta^{t-i} \nabla f^{(i)} \hat{Q}'(w^{(i)}) \tag{38}$$

Theorems D.1 and D.2 show that Assumption C.1.3 holds for this update rule under mild conditions. From this we can apply Theorem C.1 for SGD with momentum to obtain Theorem D.3, a result similar to Theorem 5.1.

**Theorem D.1.** *Define $g^{(t)}$ by Equation 38. Suppose that Assumption 5.1.1 holds. Further suppose that each $\nabla f^{(t)}$ is bounded by*

$$|\nabla f^{(t)}| < \frac{g_+}{L_+(1-\beta^{t+1})}. \tag{39}$$

*Then*

$$|g^{(t)}(\nabla f^{(0)}\hat{Q}'(w^{(0)}),\ldots,\nabla f^{(t)}\hat{Q}'(w^{(t)}),\eta)| < \eta g_+$$

*Proof.* By the triangle inequality and Assumption 5.1.1, we have

$$|g^{(t)}(\nabla f^{(0)}\hat{Q}'(w^{(0)}),\ldots,\nabla f^{(t)}\hat{Q}'(w^{(t)}),\eta)| < \eta L_+(1-\beta)\sum_{i=0}^{t}\beta^{t-i}|\nabla f^{(i)}|.$$

Now applying the bound given in Equation 39, we have

$$|g^{(t)}(\nabla f^{(0)}\hat{Q}'(w^{(0)}),\ldots,\nabla f^{(t)}\hat{Q}'(w^{(t)}),\eta)| < \eta g_+\frac{1-\beta}{1-\beta^{t+1}}\sum_{i=0}^{t}\beta^{t-i}.$$

Since

$$\sum_{i=0}^{t}\beta^{t-i} = \frac{1-\beta^{t+1}}{1-\beta}$$

for all $\beta < 1$, we have

$$|g^{(t)}(\nabla f^{(0)}\hat{Q}'(w^{(0)}),\ldots,\nabla f^{(t)}\hat{Q}'(w^{(t)}),\eta)| < \eta g_+ \tag{40}$$

as desired. $\square$

**Theorem D.2.** *Define $g^{(t)}$ by Equation 38. Suppose that*

*D.2.1 $0 < L_- \le \hat{Q}'(w)$ for all $w$*

*D.2.2 $\hat{Q}'(w)$ is $L'$-Lipschitz*

*D.2.3 For each $t$, Each $g^{(t)}$ is bounded by $|w^{(t+1)} - w^{(t)}| < \eta g_+$.*

*Then for each $t$, we have*

$$\left|\frac{g^{(t)}(\nabla f^{(0)}\hat{Q}'(w^{(0)}),\ldots,\nabla f^{(t)}\hat{Q}'(w^{(t)}),\eta)}{\hat{Q}'(w^{(t)})} - g^{(t)}(\nabla f^{(0)},\ldots,\nabla f^{(t)},\eta)\right| = O(\eta^2). \tag{41}$$

*so that Assumption C.1.3 holds with $c(\eta) = \eta^2$.*

*Proof.* We have by Equation 38

$$\frac{g^{(t)}(\nabla f^{(0)}\hat{Q}'(w^{(0)}),\ldots,\nabla f^{(t)}\hat{Q}'(w^{(t)}),\eta)}{\hat{Q}'(w^{(t)})} = -\eta(1-\beta)\sum_{i=0}^{t}\beta^{t-i}\nabla f^{(i)}\frac{\hat{Q}'(w^{(i)})}{\hat{Q}'(w^{(t)})}. \tag{42}$$

We would like to show that for each $i$,

$$\beta^{t-i}\frac{\hat{Q}'(w^{(i)})}{\hat{Q}'(w^{(t)})} = \beta^{t-i}(1 + O(\eta))$$

since then we would have

$$\left| \frac{g^{(t)}(\nabla f^{(0)} \hat{Q}'(w^{(0)}), \ldots, \nabla f^{(t)} \hat{Q}'(w^{(t)}), \eta)}{\hat{Q}'(w^{(t)})} - g^{(t)}(\nabla f^{(0)}, \ldots, \nabla f^{(t)}, \eta) \right| = \tag{43}$$

$$\left| -\eta(1-\beta) \sum_{i=0}^{t} \beta^{t-i} \nabla f^{(i)} \frac{\hat{Q}'(w^{(i)})}{\hat{Q}'(w^{(t)})} + \eta(1-\beta) \sum_{i=0}^{t} \beta^{t-i} \nabla f^{(i)} \right| = \tag{44}$$

$$\left| -\eta(1-\beta) \sum_{i=0}^{t} \beta^{t-i} \nabla f^{(i)} (1 + O(\eta)) + \eta(1-\beta) \sum_{i=0}^{t} \beta^{t-i} \nabla f^{(i)} \right| = O(\eta^2) \tag{45}$$

$$\tag{46}$$

The first step is to note that $\log(\hat{Q}')$ is Lipschitz with Lipschitz constant $L'/L_-$. To see this, first note that $\log(x)$ is $1/L_-$-Lipschitz on the range $[L_-, \infty]$. Then by Assumptions D.2.1 and D.2.2 and the fact that the composition of Lipschitz functions is Lipschitz with the product constant, we have

$$|\log(\hat{Q}'(w)) - \log(\hat{Q}'(v))| \leq \frac{L'}{L_-} |w - v|$$

which is our desired Lipschitz property. Making use of this property, Assumption D.2.3, and Equation 2, we have

$$|\log(\hat{Q}'(w^{(i)})) - \log(\hat{Q}'(w^{(t)}))| \leq \frac{L'}{L_-} |w^{(i)} - w^{(t)}| \tag{47}$$

$$= \frac{L'}{L_-} \left| \sum_{j=i}^{t-1} w^{(i)} - w^{(i+1)} \right| \tag{48}$$

$$\leq \frac{L'}{L_-} \sum_{j=i}^{t-1} \left| w^{(i)} - w^{(i+1)} \right| \tag{49}$$

$$\leq \eta \frac{L'}{L_-} (t-i) g_+. \tag{50}$$

Solving for the quotient $\hat{Q}'(w^{(i)})/\hat{Q}'(w^{(t)})$, we have

$$-\eta L'(t-i) g_+/L_- \leq \log(\hat{Q}'(w^{(i)})) - \log(\hat{Q}'(w^{(t)})) \leq \eta L'(t-i) g_+/L_-$$

$$\exp(-\eta L'(t-i) g_+/L_-) \leq \frac{\hat{Q}'(w^{(i)})}{\hat{Q}'(w^{(t)})} \leq \exp(\eta L'(t-i) g_+/L_-)$$

$$\beta^{-\eta L'(t-i) g_+/(\log(\beta) L_-)} \leq \frac{\hat{Q}'(w^{(i)})}{\hat{Q}'(w^{(t)})} \leq \beta^{\eta L'(t-i) g_+/(\log(\beta) L_-)}$$

Thus we have shown that

$$\frac{\hat{Q}'(w^{(i)})}{\hat{Q}'(w^{(t)})} = \left( \frac{\beta_{t,i}}{\beta} \right)^{t-i}$$

where

$$\beta_{t,i} = \beta + O(\eta).$$

Therefore we have

$$\beta^{t-i} \frac{\hat{Q}'(w^{(i)})}{\hat{Q}'(w^{(t)})} = \beta_{t,i}^{t-i} = (\beta + O(\eta))^{t-i} = \beta^{t-i}(1 + O(\eta)),$$

as desired. The final equality holds since $(\beta + O(\eta))^{t-i}$ is a polynomial in $\beta$ and $O(\eta)$, which can be computed by expanding the product. Each term in the resulting sum is either $\beta^{t-i}$, $O(\eta)$, or $o(\eta)$. $\square$

We now have all that we need to the following analog of Theorem 5.1 for gradient descent with momentum.

**Theorem D.3.** *Suppose that $E^{(t)}$ is defined by Equation 14, for $\hat{Q}$-net and $STE$-net with gradient estimators, learning rates, and initial weights given by Table 1. Suppose that Assumptions 5.1.1 and 5.1.2 hold and the model weights are updated according to Equation 2, where $g^{(t)}$ is defined by Equation 38. In addition, suppose that each $\nabla f_{\hat{Q}}^{(t)}$ is bounded by Equation 39. Then we have*

$$E^{(t+1)} \leq E^{(t)} + \alpha\eta \left| (1-\beta)\sum_{i=0}^{t}\beta^{t-i}(\nabla f_{\hat{Q}}^{(i)} - \nabla f_{STE}^{(t)}) \right| + \frac{L'}{2}\cdot\left(\frac{\eta g_+}{L_-}\right)^2 + O(\eta^2) \quad (51)$$

*Proof.* Assumption C.1.3 holds by Theorem D.2 with $c(\eta) = \eta^2$, so that Theorem C.1 holds. Note that Assumption D.2.3 holds by a combination of Theorem D.1 and Equation 2. We can now obtain Equation 51 from Equation 17 by simplifying terms and applying the appropriate bounds:

$$E^{(t+1)} \leq E^{(t)} + \left| \alpha g^{(t)}(\alpha\nabla f_{\hat{Q}}^{(0)}, \ldots, \nabla f_{\hat{Q}}^{(t)}, \eta) - g^{(t)}(\nabla f_{STE}^{(0)}, \ldots, \nabla f_{STE}^{(t)}, \alpha\eta) \right| + \quad (52)$$

$$\frac{L'}{2}\cdot\left(\frac{g^{(t)}(\nabla f_{\hat{Q}}^{(0)}\hat{Q}'(w^{(0)}), \ldots, \nabla f_{\hat{Q}}^{(t)}\hat{Q}'(w^{(t)}), \eta)}{L_-}\right)^2 + O(c(\eta)) \quad (53)$$

$$\leq E^{(t)} + \left| -\alpha\eta(1-\beta)\sum_{i=0}^{t}\beta^{t-i}\nabla f_{\hat{Q}}^{(i)} + \alpha\eta(1-\beta)\sum_{i=0}^{t}\beta^{t-i}\nabla f_{STE}^{(i)} \right| + \quad (54)$$

$$\frac{L'}{2}\cdot\left(\frac{\eta g_+}{L_-}\right)^2 + O(\eta^2) \quad (55)$$

$$= E^{(t)} + \alpha\eta \left| (1-\beta)\sum_{i=0}^{t}\beta^{t-i}(\nabla f_{\hat{Q}}^{(i)} - \nabla f_{STE}^{(t)}) \right| + \frac{L'}{2}\cdot\left(\frac{\eta g_+}{L_-}\right)^2 + O(\eta^2). \quad (56)$$

$\square$

# E  ADAM

In this Appendix we prove Theorem 5.2 in a manner similar to the proofs given in Appendix C. The weight update function for the Adam optimizer is defined by

$$m_t = \beta_1 m_{t-1} + (1-\beta_1)\nabla f^{(t)}\hat{Q}'(w^{(t)}) \quad (57)$$

$$v_t = \beta_2 v_{t-1} + (1-\beta_2)(\nabla f^{(t)}\hat{Q}'(w^{(t)}))^2 \quad (58)$$

$$\hat{m}_t = m_t/(1-\beta_1^t) \quad (59)$$

$$\hat{v}_t = v_t/(1-\beta_2^t) \quad (60)$$

$$g^{(t)}(\nabla f^{(0)}\hat{Q}'(w^{(0)}), \ldots, \nabla f^{(t)}\hat{Q}'(w^{(t)}), \eta) = -\eta\hat{m}_t/\left(\sqrt{\hat{v}_t} + \epsilon\right) \quad (61)$$

where $\beta_1, \beta_2 \in [0,1)$ are hyperparameters and $\epsilon$ is a small constant.

We will first state and prove Theorem E.1, ageneral-purpose precursor to Theorem 5.2 that applies to a large class of adaptive learning rate optimizers. Then we will borrow work from the proof of Theorem D.2 to specify this result for the Adam optimizer and prove Theorem 5.2.

Throughout this section, we will follow Kingma and Ba (2014) and assume for the sake of mathematical argument that the constant $\epsilon$ in Equation 61 is zero.

**Theorem E.1.** *Suppose that*

$$E^{(t)} := \left| w_{\hat{Q}}^{(t)} - w_{STE}^{(t)} \right| \quad (62)$$

*is the alignment error for $\hat{Q}$-net and $STE$-net with gradient estimators, learning rates, and initial weights given by Table 2. Suppose that the model weights are updated according to Equation 2 for some function $g^{(t)}$. In addition, suppose that*

*E.1.3 For each t, the quantity*

$$\left| g^{(t)}(\nabla f_{\hat{Q}}^{(0)} \hat{Q}'(w^{(0)}), \ldots, \nabla f_{\hat{Q}}^{(t)} \hat{Q}'(w^{(t)}), \eta) - g^{(t)}(\nabla f_{\hat{Q}}^{(0)}, \ldots, \nabla f_{\hat{Q}}^{(t)}, \eta) \right| = O(c(\eta)). \tag{63}$$

*Then we have*

$$E^{(t+1)} \leq E^{(t)} + \left| g^{(t)}(\nabla f_{\hat{Q}}^{(0)}, \ldots, \nabla f_{\hat{Q}}^{(t)}, \eta) - g^{(t)}(\nabla f_{STE}^{(0)}, \ldots, \nabla f_{STE}^{(t)}, \eta) \right| + O(c(\eta)) \tag{64}$$

*Proof.* By Equation 2, we have

$$E^{(t+1)} = \left| w_{\hat{Q}}^{(t+1)} - w_{STE}^{(t+1)} \right| \tag{65}$$

$$= \left| w_{\hat{Q}}^{(t)} + g^{(t)}(\nabla f_{\hat{Q}}^{(0)} \hat{Q}'(w^{(0)}), \ldots, \nabla f_{\hat{Q}}^{(t)}, \eta) - \right. \tag{66}$$

$$\left. \left( w_{STE}^{(t)} + g^{(t)}(\nabla f_{STE}^{(0)}, \ldots, \nabla f_{STE}^{(t)}, \eta) \right) \right| \tag{67}$$

$$= \left| w_{\hat{Q}}^{(t)} + g^{(t)}(\nabla f_{\hat{Q}}^{(0)}, \ldots, \nabla f_{\hat{Q}}^{(t)}, \eta) + O(c(\eta)) - \right. \tag{68}$$

$$\left. \left( w_{STE}^{(t)} + g^{(t)}(\nabla f_{STE}^{(0)}, \ldots, \nabla f_{STE}^{(t)}, \eta) \right) \right| \tag{69}$$

$$\leq E^{(t)} + \left| g^{(t)}(\nabla f_{\hat{Q}}^{(0)}, \ldots, \nabla f_{\hat{Q}}^{(t)}, \eta) - g^{(t)}(\nabla f_{STE}^{(0)}, \ldots, \nabla f_{STE}^{(t)}, \eta) \right| + O(c(\eta)) \tag{70}$$

Here Equation 70 follows from the triangle inequality, and Equation 69 follows from Assumption C.1.3. $\qquad\square$

Now we can prove Theorem 5.2.

*Proof of Theorem 5.2.* To prove Theorem 5.2, we need to show that the assumptions of Theorem 5.2 imply the Assumption E.1.3 of Theorem E.1 with the Adam update rule defined in Equations 57-61 and $c(\eta) = \eta^2$.

We first expand Equations 57 and 58, which will allow us to express $g^{(t)}$ more explicitly as a function of the $\nabla f_{\hat{Q}}^{(i)} \hat{Q}'(w^{(i)})$:

$$m_t = (1 - \beta_1) \sum_{i=0}^{t} \beta_1^{t-i} \nabla f_{\hat{Q}}^{(i)} \hat{Q}'(w^{(i)}) \tag{71}$$

$$v_t = (1 - \beta_2) \sum_{i=0}^{t} \beta_2^{t-i} (\nabla f_{\hat{Q}}^{(i)} \hat{Q}'(w^{(i)}))^2 \tag{72}$$

$$g^{(t)}(\nabla f^{(0)} \hat{Q}'(w^{(0)}), \ldots, \nabla f^{(t)} \hat{Q}'(w^{(t)}), \eta) = -\frac{1 - \beta_1}{1 - \beta_1^t} \cdot \sqrt{\frac{1 - \beta_2^t}{1 - \beta_2}} \cdot \tag{73}$$

$$\frac{\eta \sum_{i=0}^{t} \beta_1^{t-i} \nabla f_{\hat{Q}}^{(i)} \hat{Q}'(w^{(i)})}{\sqrt{\sum_{i=0}^{t} \beta_2^{t-i} (\nabla f_{\hat{Q}}^{(i)} \hat{Q}'(w^{(i)}))^2} + \epsilon} \tag{74}$$

Clearly the two fraction terms of Equation 73 are not dependent on $\hat{Q}'$ in any way, so we need only concern ourselves with the final fraction term in Equation 74. As stated earlier, we are ignoring the $\epsilon$ term, which allows us to write the final fraction as

$$\frac{\sum_{i=0}^{t} \beta_1^{t-i} \nabla f_{\hat{Q}}^{(i)} \hat{Q}'(w^{(i)})}{\sqrt{\sum_{i=0}^{t} \beta_2^{t-i} (\nabla f_{\hat{Q}}^{(i)} \hat{Q}'(w^{(i)}))^2}} = \frac{\hat{Q}'(w^{(t)})}{\hat{Q}'(w^{(t)})} \cdot \frac{\eta \sum_{i=0}^{t} \beta_1^{t-i} \nabla f_{\hat{Q}}^{(i)} \hat{Q}'(w^{(i)})}{\sqrt{\sum_{i=0}^{t} \beta_2^{t-i} (\nabla f_{\hat{Q}}^{(i)} \hat{Q}'(w^{(i)}))^2}} \tag{75}$$

$$= \frac{\eta \sum_{i=0}^{t} \beta_1^{t-i} \nabla f_{\hat{Q}}^{(i)} \hat{Q}'(w^{(i)}) / \hat{Q}'(w^{(t)})}{\sqrt{\sum_{i=0}^{t} \beta_2^{t-i} (\nabla f_{\hat{Q}}^{(i)} \hat{Q}'(w^{(i)}) / \hat{Q}'(w^{(t)}))^2}} \tag{76}$$

We would like to apply Theorem D.2 to both the numerator and denominator of the final term in the above Equation. Assumptions D.2.1 and D.2.2 are the same as Assumptions 5.2.1 and 5.2.2, respectively. By Equation 2, we can see that Assumption D.2.3 with $g_+ = \max\{1, (1 - \beta_1)/\sqrt{(1 - \beta_2)}\}$ is an inherent property of the Adam optimizer (Kingma and Ba, 2014). Now by applying Theorem D.2 to the numerator, we have

$$\eta \sum_{i=0}^{t} \beta_1^{t-i} \nabla f_{\hat{Q}}^{(i)} \hat{Q}'(w^{(i)}) / \hat{Q}'(w^{(t)}) = \eta \sum_{i=0}^{t} \beta_1^{t-i} \nabla f^{(i)} + O(\eta^2).$$

we see that the numerator limits to $\sum_{i=0}^{t} \beta^{t-i} \nabla f^{(i)}$ as $\eta \to 0$. We can show via a very similar proof that the denominator can be approximated as

$$\sqrt{\sum_{i=0}^{t} \beta_2^{t-i} (\nabla f^{(i)})^2 + O(\eta)}.$$

The only notable differences are that we are removing an $\eta$ term, and the exponent in the bound for $\hat{Q}'(w^{(i)})/\hat{Q}'(w^{(t)})$ has an extra 2 in it, which does not affect the result. Therefore we have

$$g^{(t)}(\nabla f^{(0)} \hat{Q}'(w^{(0)}), \ldots, \nabla f^{(t)} \hat{Q}'(w^{(t)}), \eta) = -\frac{1 - \beta_1}{1 - \beta_1^t} \cdot \sqrt{\frac{1 - \beta_2^t}{1 - \beta_2}}. \tag{77}$$

$$\frac{\eta \sum_{i=0}^{t} \beta_1^{t-i} \nabla f_{\hat{Q}}^{(i)} \hat{Q}'(w^{(i)})}{\sqrt{\sum_{i=0}^{t} \beta_2^{t-i} (\nabla f_{\hat{Q}}^{(i)} \hat{Q}'(w^{(i)}))^2}} \tag{78}$$

$$= -\frac{1 - \beta_1}{1 - \beta_1^t} \cdot \sqrt{\frac{1 - \beta_2^t}{1 - \beta_2}}. \tag{79}$$

$$\frac{\eta \sum_{i=0}^{t} \beta_1^{t-i} \nabla f_{\hat{Q}}^{(i)} \hat{Q}'(w^{(i)}) + O(\eta^2)}{\sqrt{\sum_{i=0}^{t} \beta_2^{t-i} (\nabla f^{(i)})^2 + O(\eta)}} \tag{80}$$

$$\frac{\eta \sum_{i=0}^{t} \beta_1^{t-i} \nabla f_{\hat{Q}}^{(i)} \hat{Q}'(w^{(i)})}{\sqrt{\sum_{i=0}^{t} \beta_2^{t-i} (\nabla f_{\hat{Q}}^{(i)} \hat{Q}'(w^{(i)}))^2}} \tag{81}$$

$$= -\frac{1 - \beta_1}{1 - \beta_1^t} \cdot \sqrt{\frac{1 - \beta_2^t}{1 - \beta_2}}. \tag{82}$$

$$\frac{\eta \sum_{i=0}^{t} \beta_1^{t-i} \nabla f_{\hat{Q}}^{(i)} \hat{Q}'(w^{(i)})}{\sqrt{\sum_{i=0}^{t} \beta_2^{t-i} (\nabla f^{(i)})^2}} + O(\eta^2) \tag{83}$$

$$= g^{(t)}(\nabla f^{(0)}, \ldots, \nabla f^{(t)}, \eta) + O(\eta^2) \tag{84}$$

so that Assumption E.1.3 holds with $c(\eta) = \eta^2$. The only potential issue with this derivation is in the removal of the denominator $O(\eta)$ term in Equation 83. In order for this to work, we need the denominator to be nonzero. However, if the denominator is zero, then Assumption E.1.3 holds trivially. This concludes the proof.

Note: The reader may be concerned as to why the $\hat{Q}'(w^{(i)})$ terms disappeared from $g^{(t)}$ but the $\nabla f^{(i)}$ terms did not. The reason is that the $\hat{Q}'(w^{(i)})$ terms vary continuously with the latent weight, whereas the $\nabla f^{(i)}$ terms are stochastic. □

## F  LEARNING RATE SCHEDULES

**Learning rate schedules.** All of the learning algorithms described in Section 3 can make use of a learning rate schedule (Robbins and Monro, 1951; Darken et al., 1992; Li and Arora, 2019; Loshchilov

and Hutter, 2016; Smith, 2017). A learning rate schedule essentially amounts to scaling each the gradient update steps $g^{(t)}$ by a pre-determined positive number $\eta_t$. In this case, the initial learning rate $\eta$ acts as a scale on the entire learning rate schedule.

Theorems C.1 and E.1 are general-purpose tools for proving results like Theorems 5.1 and 5.2 for non-adaptive learning rate optimizers and adaptive learning rate optimizers, respectively. Up until this point, we have only focused on fixed learning rate schedules, and here we describe how the theorems be applied to general learning rate schedules.

As stated in Section 3, a learning rate schedule applies a pre-determined scale $\eta_t$ to each of the gradient update steps $g^{(t)}$, which can effectively be absored into the $\nabla f^{(t)}$ terms for non-adaptive optimizers. This does not affect Assumptions 5.1.1, 5.1.2, 5.2.1, or 5.2.2 in any way. It may affect the bounds on $\nabla f_{\hat{Q}}^{(t)}$ in Theorem D.3, but this would simply require a different value of $g_+$.

Thus we can confidently generalize our main results to gradient update rules that take advantage of learning rate schedules.

## G ON NONPOSITIVE GRADIENT ESTIMATORS

Here we describe the statements we can make that bear relation to Theorems 5.1 and 5.2 for gradient estimators that break the lower bound conditions in Assumptions 5.1.1 and 5.2.1.

**The common case for nonpositive gradient estimators.** Assumptions 5.1.1 and 5.2.1 are most commonly broken when $\hat{Q}'$, like the PWL estimator (See Section 2), is positive on some range $[w_{min}, w_{max}]$ and zero outside of this range. The behavior of these gradient estimators cannot be mimicked by any model that uses the STE, since the latent weight can reach a point where it no longer receives updates from gradients. However, this behavior *can* be mimicked by a model that uses PWL estimator. If we set

$$\tilde{w}_{min} := M(w_{min}) \tag{85}$$
$$\tilde{w}_{max} := M(w_{max}), \tag{86}$$

then Theorems 5.1 and 5.2 clearly apply after replacing the STE with $PWL_{\tilde{w}_{min}, \tilde{w}_{max}}$ (for SGD), $PWL_{w_{min}, w_{max}}$ (for Adam), whenever $w_{\hat{Q}}^{(t)}$ and $w_{STE}^{(t)}$ are in the representable range. Technically, $M(w_{\hat{Q}}^{(0)})$ is only defined when $w_{\hat{Q}}^{(0)} \in [w_{min}, w_{max}]$, but we can ignore this case under the assumption that no practitioner would initialize a weight to be untrainable. There are two remaining cases to consider. The first is where $w_{\hat{Q}}^{(t)}$ and $w_{STE}^{(t)}$ both lay outside of the representable range, in which case neither weight can move and there is no risk of increasing $E^{(t)}$. The second is where only one lies in this range, and one weight is "trapped" while the other is "free". This is unlikely to happen due to the bounds on $E^{(t)}$, but it could technically lead to high weight alignment errors.

**Negative gradient estimators.** The other way that the lower bound in Assumption 5.1.1 can be broken is if $\hat{Q}(w)$ is actually negative for some range of values of $w$. There is some work (Darabi et al., 2018; Xu et al., 2021) that proposes gradient estimators with negative derivatives, but most choose a nonnegative derivative to align with the nondecreasing behavior of the quantizer function. In the cases with negative $\hat{Q}'$ values, slightly modified versions of Theorems 5.1 and 5.2 apply on the negative ranges, where the gradient estimator of $STE$-net is the negative of the STE. Since this is a rare choice for QAT, we do not provide the details here.

Thus almost all common gradient estimators can be replaced with the STE or a PWL estimator.

## H CALCULATING CONSTANTS IN THEOREM 5.1

Many gradient estimators take the form

$$\hat{Q}(w) = \tanh(k \cdot (w - a) + a$$

for $w$ in the representable range, and $a$ is the center of the quantization bin $w$ is in. This is the case for Gong et al. (2019) and Pei et al. (2023), hence our choice of the gradient estimator from Pei et al.

(2023) for the experiments. This is also very similar to the gradient estimator used in Yang et al. (2019).

Given this definition of $\hat{Q}(w)$, we want to provide lower and upper bounds on the first and second derivatives of $Q$ on the interval $[-\Delta/2, \Delta/2]$ with $a = 0$. First note that we have

$$\hat{Q}'(w) = \frac{k}{\cosh^2(kw)}$$

This obtains a maximum value at $w = 1$, and a minimum value at $\pm\Delta/2$, so that $L_+ = k$ and $L_- = k/\cosh^2(k\Delta/2)$.

$$\hat{Q}''(w) = -2k^2 \frac{\tanh(kw)}{\cosh^2(kw)}$$

This obtains its maximum values at

$$w = \pm\frac{1}{2k}\log(2 + \sqrt{3})$$

and is strictly decreasing on the interval between these points. Since a bound on $|\hat{Q}''(w)|$ is a Lipschitz constant for $\hat{Q}'$, $L'$ is given by

$$2k^2 \frac{\tanh(kw)}{\cosh^2(kw)}$$

where

$$w = \min\left(\Delta/2, \frac{1}{2k}\log(2 + \sqrt{3})\right)$$

In Pei et al. (2023), $k$ is set to to 8, 6, 4, and 2 for 8, 4, 3, and 2-bit quantization. They initialize $\Delta$ to $2/(2^b - 1)$ where $b$ is the number of bits used for quantization. This gives us the following values for $L'L_+/2L_-^2$: 0.25 (8 bits), 2.66 (4 bits), 2.82 (3 bits), 1.77 (2 bits). These values are small relative to standard values of $1/\eta$, where $\eta$ is the learning rate.

For Gong et al. (2019), the quantizer is parametrized by a value $\alpha$ defined by

$$\alpha = 1 - \tanh(k\Delta/2).$$

This gives us convenient formulas:

$$\tanh(k\Delta/2) = 1 - \alpha$$

$$\frac{1}{\cosh(k\Delta/2)^2} = 1 - (1 - \alpha)^2 = 2\alpha - \alpha^2$$

$$\frac{\tanh(k\Delta/2)}{\cosh(k\Delta/2)^2} = (1 - \alpha)(2\alpha - \alpha^2)$$

$$\frac{L_+}{L_-} = \frac{1}{2\alpha - \alpha^2}$$

$$\frac{L'L_+}{L_-^2} \le \frac{1 - \alpha}{2\alpha - \alpha^2}$$

The constant of interest is then given by

$$\frac{L'L_+}{L_-^2} \le \frac{1 - \alpha}{2\alpha - \alpha^2}$$

During training in Gong et al. (2019), $\alpha$ is varied for weight quantizers between 0.11 and 0.25, giving us

$$\frac{L'L_+}{L_-^2} \in [1.71, 4.28].$$

These values are again small relative to $1/\eta$.

# I  FUTURE WORK

**Extension to other gradient estimators:** Our results can be adapted to noncyclical and nonuniform gradient estimators. We addressed the common case here, and avoided the more general case due to the notational complexity required.

**Extensive study of learning rate and weight alignment error throughout training**: Our experiments could be extended to give more detailed empirical data on the relationship between learning rates and weight alignment error. For example, weight alignment error could be collected throughout training, for many different models/datasets and for many different learning rates.

# J  VISUAL FOR WEIGHT ALIGNMENT

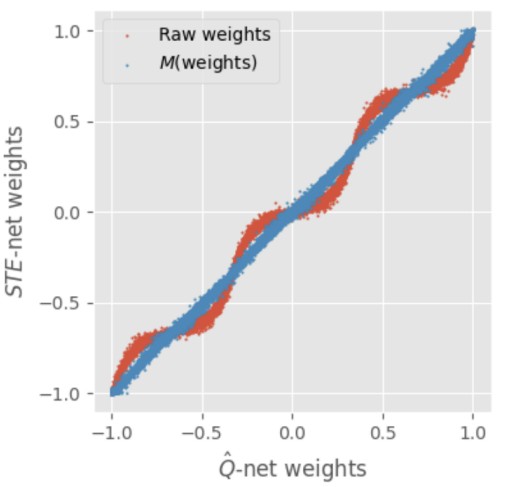 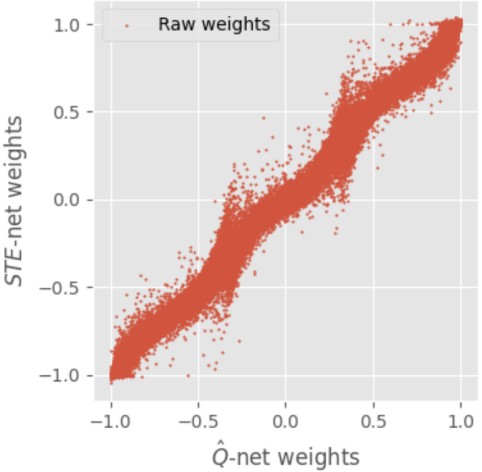

(a) $\hat{Q}$-net weights vs $STE$-net weights for MNIST convolutional model at the conclusion of training for default SGD.

(b) $\hat{Q}$-net weights vs $STE$-net weights at the conclusion of training *without* re-initializing $STE$-net weights.

Figure 3: Comparison of model weights at the end of training for MNIST model.

We visualize the alignment of weights between $STE$-net and $\hat{Q}$-net in Figure 3a. We can see that after applying $M$, the weights of the two models are very closely aligned at the end of training. Furthermore, if we do not apply the appropriate weight initialization to $STE$-net, the weights are no longer aligned at the end of training. This is shown in Figure 3b.

