# OpenReview forum: "Custom Gradient Estimators are Straight-Through Estimators in Disguise"
_ICLR.cc/2025/Conference — Submitted to ICLR 2025_

### Official Review · Reviewer_KqAp · 2024-10-27

**Soundness:** 3
**Presentation:** 3
**Contribution:** 2
**Rating:** 5
**Confidence:** 3

**Summary:**

The authors examine how different gradient estimators affect quantization-aware training. They theoretically demonstrate that, in the limit of small learning rates and with minor adjustments to the initialization and the learning rate magnitude, most gradient estimators yield equivalent weight movements. Consequently, they suggest that the Straight-Through Estimator, treating the gradient as if no quantization occurred in the backward pass, performs as well as any other more sophisticated alternative. Their theoretical claims are complemented by an empirical investigation.

**Strengths:**

Equations (6) and (7) provide a rigorous theoretical contribution regarding the small difference in weight movement for different gradient estimators.

The presentation and writing style is very clear with helpful intuition such as the analogy of the "funhouse mirror". Additionally, the learning rate tweak in the experiment provides a practical comparison for the magnitude of the differences.

**Weaknesses:**

The authors themselves acknowledge in Section 8 that many publications introduce more than just a new gradient estimator, raising questions about the broader practical impact of their findings. Novel gradient estimators are often proposed with other techniques to enable any benefits. As a result, the findings may have limited applicability beyond specific configurations.

It is somewhat difficult to draw clear conclusions about practical applications from the experiments. While the learning rate tweak provides a useful comparison, the differences in Tables 4 and 5 are challenging to interpret, particularly without comparisons across different initializations. A convincing experiment could be, to find a custom gradient estimator in the literature, which has been shown to improve validation accuracy over STE, replicate the results and then demonstrate that both perform equally with proper adjustments. The authors do not provide such a comparison, raising questions about whether the custom estimator was applied correctly in their experiments or if its potential advantages were overlooked.


Minor comments:

line 245: Typo: $Q(w_{\hat{Q}}^{(t)})=Q(w_{\hat{Q}}^{(t)})$

lines 329-340: Equations (8) to (12): "$\nabla$" missing before $f^{(t)}_{STE}$

lines 502-504: Missing figure number for Figure 3

**Questions:**

Lines 400-404: Why was the initial 10% of training for the ImageNet-ResNet setup kept identical? What would the results be without this measure?

Lines 527-528: Regarding the statement that high learning rates might be the reason the equivalence is not observed in other studies, the authors write, "we expect that this counter-argument will not stand the test of time, since by our main results, the higher learning rate masks the fact that models with novel $\hat{Q}$ and the STE are still approximating the same process." Could you elaborate on what you mean by "will not stand the test of time" given that equations (6) and (7) indicate learning-rate-squared errors? It seems that higher learning rates would increase these errors. Why should these differences not enable advantages for novel $\hat{Q}$?


Lines 530-539: Can the authors comment on the potential implications of their results for the studies mentioned in the last paragraph, which propose additional innovations alongside novel gradient estimators?

---

> ### Author Response · Authors · 2024-11-25
> **Our results explain why gradient estimators are never introduced as standalone contributions**
>
> ## General Response
> Thank you for your thoughtful review and constructive feedback. We are pleased the see that the reviewer recognized the theoretical strength of the paper and the empirical support for our findings. Below, we address your comments and questions in detail.
>
> ---
>
> ## Weaknesses
>
> 1. **Practical Impact and Broader Applicability**
>    Interestingly, gradient estimators are never proposed in the literature as standalone contributions that improve performance. This point is highlighted in the final paragraph of the paper, and we believe that our work provides an explanation for this!
>
> 2. **Suggested Experiment**
>    The experiment you proposed would indeed be convincing. However, there are no existing studies that directly contradict our claims by demonstrating a custom gradient estimator outperforming the STE under the conditions we analyzed. As a result, such an experiment cannot be conducted in a meaningful way within the scope of this work.
>
> ---
>
> ## Minor Comments
>
> 1. **Typographical Errors**
>    - Thank you for identifying the typo on Line 245. We have corrected this to:
>      $$
>      Q(w_{\hat{Q}}^{(t)}) = Q(w_{\text{STE}}^{(t)})
>      $$
>    - We have also corrected the missing $\nabla$ terms in Equations (8)–(12) and the figure reference for Figure 3 in Lines 502–504.
>
> ---
>
> ## Questions
>
>
> 1. **Initial 10% of Training with Identical Gradient Estimators**
>    Gradient movements are very large at the beginning of training, leading to correspondingly large error terms in the theorem. To mitigate this, we used the same gradient estimator for the first 10% of training. This allowed us to focus on the remainder of the training process, where precise, small weight movements dominate. Without this measure, the weights would diverge more quickly in the early epochs, making it harder to isolate the small differences studied later in training.
>
> 2. **Counter-Argument Regarding High Learning Rates**
>    You raise a valid concern regarding the influence of high learning rates. When we state that this counter-argument "will not stand the test of time," we mean that if large learning rates are the sole justification for custom gradient estimators, their value relies on a second-order error term in a Taylor approximation. Our results show that this term is very small and not as well-understood as others may have thought, making it unlikely to consistently improve training outcomes. While large learning rates do increase these errors ($O(\eta^2)$), we believe this effect is not systematic or predictable enough to provide a consistent advantage for novel estimators.
>
> 3. **Implications for Studies Combining Gradient Estimators with Additional Innovations**
>    We address this question in the third paragraph of Appendix B. Here, we discuss how gradient estimation techniques can often be reduced to well-known training recipe adjustments, such as learning rate scaling or weight initialization changes. These insights suggest that the purported benefits of novel estimators may often stem from these auxiliary adjustments rather than the gradient estimation method itself.
>
> ---
>
> We hope these clarifications address your concerns and improve the presentation of our work. Thank you again for your valuable feedback.

---

> > ### Comment · Reviewer_KqAp · 2024-11-27
> > **Response to Authors’ Rebuttal**
> >
> > I thank authors for their detailed responses to the review comments. While some of their clarifications are helpful, a few points remain unclear or insufficiently addressed.
> >
> > ---
> >
> > ### **High Learning Rate Errors**
> > The clarification regarding high learning rate effects acknowledges that $\mathcal{O}(\eta^2)$ errors are significant in the initial 10% of training, which conflicts with the claim in response to Question 2 that these terms are "very small." While it may be true that such errors diminish as training progresses, this does not sufficiently substantiate the statement that these errors are neither systematic nor predictable enough to provide a consistent advantage. Without further theoretical or experimental evidence, this remains a speculative interpretation.
> >
> > ---
> >
> > ### **Gradient Estimators with Additional Innovations**
> > The response that "gradient estimators are never proposed as standalone contributions" is noted, however, in my view, it does not yet convincingly address the broader practical implications in question. While the theoretical results may explain why standalone gradient estimators are rare, this explanation does not resolve the issue of applicability.
> >
> > Specifically, the third paragraph in Appendix B appears to oversimplify the effects of the gradient estimator from Qin et al. (2020):
> >
> > "*For example, Qin et al. (2020) proposes a schedule for a tanh-based gradient estimator to gradually approach a sign function throughout training. Since they use SGD in their experiments, we can think of each update to sharpen the gradient estimator as an effective "shifting" of the weights according to the function defined in Equation 4. This particular shift will push most weights away from 0, which has an effect similar to slowing down the learning rate. Thus this adaptive gradient estimation technique is similar to a standard learning rate decay schedule.*"
> >
> > I believe that the effects of the gradient estimator schedule result in a more complex learning rate schedule influenced by weight magnitudes. It would most likely be similar to decreasing the learning rate for weights with an absolute value above a certain evolving threshold that decreases over time, and initially increasing the learning rate for weights below that threshold. Crucially, this appears to be a more complex mechanism than the simple decay mechanism proposed by the authors. This added complexity, in my view, undermines the claim in the manuscript that practitioners can simply rely on the straight-through estimator as a default and disregard more complicated gradient estimators.
> >
> > Furthermore, the lack of comparisons with custom gradient estimators in setups proposed in the literature remains a significant limitation. While the authors argue that no studies directly contradict their claims, this does not diminish the value of replicating results from such studies and demonstrating equivalence under properly controlled conditions. Such a comparison would, in my view, strengthen their claims and provide clearer guidance for practitioners, particularly in relation to the statement that Adam requires no modifications to weights or learning rate schedules at all.

---

> > > ### Author Response · Authors · 2024-11-28
> > > **Requests are great ideas for future work**
> > >
> > > These are highly astute comments- the reviewer clearly understands the paper.
> > >
> > > Our goal in this paper was to show in a novel way how the effect of gradient estimators can in certain scenarios be reinterpreted as effects on the learning rate and weight initialization, and prove in a highly generic fashion how this relationship works for different optimizers. This opens the door to a new way of thinking about gradient estimators, which we intend to explore further in future work motivated by the reviewer's comments.
> > >
> > > The comments on learning rates are appropriate, and for this reason we have qualified our claims on the practical impacts of our work. Specifically, we now clearly say on page 1 that the equivalence holds in the low learning rate regime. This adjustment ensures that our claims are aligned with the specific conditions analyzed in our paper.
> > >
> > > The reviewers' observation that gradient estimator schedules like those in Qin et al. (2020) effectively have a complex influence on learning rates is certainly true, and an extension of the ideas proposed in this paper. We did not claim that the effect was the same as a common learning rate schedule- we merely said that it was "similar", and applied to "most weights". Furthermore, the suggested experiments would be a great avenue for future work.

---

> > > > ### Comment · Reviewer_KqAp · 2024-11-29
> > > >
> > > > I appreciate the authors' thoughtful clarifications and the proposed adjustments to emphasize the low learning rate regime. The theoretical contributions are valuable and provide a novel perspective on gradient estimators. However, in my view, the lack of a direct comparison with custom gradient estimators in the settings where they were originally proposed limits the practical impact of the claims. This missing experimental validation leaves uncertainty about their broader applicability and weakens broader claims, such as the suggestion that practitioners can confidently rely on the Straight-Through Estimator as a default. I intend to maintain my original score.

---

### Official Review · Reviewer_XPRQ · 2024-10-29

**Soundness:** 3
**Presentation:** 3
**Contribution:** 3
**Rating:** 6
**Confidence:** 3

**Summary:**

This paper presents a theoretical analysis of gradient estimators used in quantization-aware training. The authors show that in the case of quantized weights (but full precision activations) many extant gradient estimators for the quantization operation are approximately equivalent, if the learning rate and weight initialization are adjusted, and the learning rate is small. They then verify empirically their theoretical results, on image classification benchmarks, demonstrating that models trained with different gradient estimators indeed show high weight agreement and similar accuracies.

**Strengths:**

- The theoretical insights are interesting, unexpected and (to my knowledge) novel. They offer better understanding and insight into how gradient estimators work, which appeals to me.
- The paper is generally well written and easy to read. I appreciate how the authors lead their result with an intuitive explanation and illustrative graphic. This makes the following theory much easier to intuit.
- The experiments shown in the paper provide good evidence for the theoretical results.

**Weaknesses:**

Major

1. The claims relating to practical impact feel overstated ("practitioners can now confidently choose the STE"). The problem setting that the authors explore (full precision activations, quantized weights, uniform fixed point quantization, small learning rate) is rather specific, and practitioners may be interested in quantized activations, or low-precision floating point or larger learning rates etc. I would prefer if the authors tempered their claims.
2. The experiments, although they demonstrate the theory well, are limited. They do not show finetuning from a pretrained full precision checkpoint, as is common for QAT in practice (I would expect this setting to match well with the theory since 1. QAT finetuning is done with a lower learning rate typically and 2. the gradient norm is likely to be low after initializing from a pretrained model). They do not show results other than 2-bit weight quantization even though they say results are similar. They do not show the practical limits of their theory, e.g. how weight alignment degrades/evolves over training or how much the learning rate needs to be increased for the error terms to start having a large impact.

Minor
1. Presentation could be improved in a number of ways.
    1. Use of booktabs for tables. Place all table captions above the tables.
    1. Tables are hard to parse when skimming -- would benefit from more descriptive captions/grouping table 3 with 4
    2. All quotation marks are incorrectly rendered by LaTeX.
    2. Fig. 3 would look better with the bins.
2. The choice of training recipes are not explained -- it is unclear why the first 10 epochs are done using the same gradient estimator. Is it because the gradient norm is too high at the start of training resulting in the weights quickly diverging?
3. I think the point made in line 305 should be made more prominent. I think it is quite important that the reader is made aware that the gradient error is small/zero since Q-net and STE net will quantize to similar weights.

**Questions:**

See weaknesses

---

> ### Author Response · Authors · 2024-11-25
>
> We thank the reviewer for their thoughtful response and recognition of the paper's novelty.
> This paper offers a deeper and more fundamental understanding of how gradient estimators
> work, and we are glad the the author appreciates this.
>
> ## Weaknesses
>
> ### Major
>
> 1. **Overstated Claims on Practical Impact**
>
>    This is a reasonable critique, and we have tempered our claims on the practical impacts of our work. Specifically, we now clearly state the assumptions required for the equivalence between custom gradient estimators and the STE to hold. This adjustment ensures that our claims are aligned with the specific conditions analyzed in our paper.
>
> 2. **Experiment Limitations**
>
>    We considered conducting fine-tuning experiments but chose not to include them as they represent a less challenging scenario. During fine-tuning, both gradients and learning rates are typically small, making the weight alignment described in our theorems easier to achieve.
>
> We acknowledge the limitation that the practical limits of our theory, such as how weight alignment degrades over training or how large learning rates impact the error terms, are not explicitly explored in our experiments. This is an area for future work, and we have noted it in Appendix I.
>
> Unfortunately the data on other bit-widths is no longer available to us due to recent employment changes
> among the authors.
>
> ---
>
> ### Minor
>
> 1. **Presentation Improvements**
>    We have made the recommended presentation adjustments, including:
>    - Using booktabs for tables and placing all table captions above the tables.
>    - Correcting the rendering of quotation marks in LaTeX.
>    - Figure 3 has been moved to Appendix J to make room for other changes. The grid in the background gives the quantization bins.
>
>    These changes significantly improve the readability and presentation of the paper!
>
> 2. **Pre-training period**
>    Gradient movements are very large at the beginning of training, which makes the error terms in the theorem correspondingly large. To mitigate this, we used the same gradient estimator for the first 10 epochs. This allows us to focus on the longer portion of the training process, where precise and small weight movements dominate.
>
> 3. **Prominence of Line 305's Point**
>    This is an excellent suggestion. We have italicized the moved the claim from Line 305.

---

> > ### Comment · Reviewer_XPRQ · 2024-11-25
> >
> > Dear Authors,
> >
> > Thank you for addressing my review. It does not appear that the submission PDF has been updated unfortunately so I cannot evaluate any of the updates.

---

> > > ### Author Response · Authors · 2024-11-25
> > >
> > > Thank you for reminding us! We have uploaded an updated pdf.

---

> > > > ### Comment · Reviewer_XPRQ · 2024-11-25
> > > >
> > > > I would like to thank the authors for the update. Although the authors have improved the paper according to the points raised in my review, the limited number of experiments remains a weakness. I understand the authors' circumstances may preclude additional experiments, but I can only evaluate the paper as it is.
> > > >
> > > > I will thus keep my score unchanged.

---

### Official Review · Reviewer_oPpR · 2024-10-30

**Soundness:** 2
**Presentation:** 2
**Contribution:** 2
**Rating:** 5
**Confidence:** 4

**Summary:**

This paper studies behavior of gradient estimators, including straight through
estimator (STE), for weight quantization. It is shown that a large class of
weight gradient estimators is approximately equivalent to the STE during training using SGD and Adam.

**Strengths:**

1. The paper is overall well presented.
2. The concept of mirror effect is interesting.

**Weaknesses:**

A primary concern is that the key claims and several major concepts lack mathematical rigor. Additionally, the main theoretical results provided are too limited to substantiate the claims:

1. Contribution 1 states that '... all nonzero weight gradient estimators lead to approximately equivalent weight movement for non-adaptive learning rate optimizers ...'. However, the term 'approximately equivalent weight movement' lacks a precise mathematical definition. It would be helpful to formalize this concept, perhaps by specifying the conditions under which these movements are considered 'approximately equivalent.'

2. According to Section 6.2, 'approximately equivalent weight movement' appears to refer to high 'Quantized Weight Agreement' or a small 'Normalized Weight Alignment Error ($\bar{E}$)'. Again, these metrics require explicit mathematical expressions for each. Additionally, this interpretation is not fully supported by the main theoretical results (e.g., Theorem 5.1), which only derive the increment in weight alignment error between two consecutive iterations, rather than a direct measure of agreement or alignment over the entire optimization trajectory.

3. For the error bounds in Eq. (6) and (7), there is insufficient justification for why the gradient error terms should be small, nor any clear indication of how small these terms are. It is insufficient to merely claim that a term is 'small' and then disregard it. These errors could accumulate significantly over iterations, potentially undermining the main conclusions.

4. The use of mathematical notation is poor, which possibly lead to incorrect derivations. For example:
   - The Euclidean norm should be denoted by $\|| \cdot \||$ rather than $\| \cdot \|$ as in Eq. (5)-(12) and other instances.
   - It should be explicitly stated that  $Q$ and $M$ are applied *element-wise* to the weight vector. Additionally, it would be preferable to use bold letters to represent vectors and to distinguish them from scalars.
   - In Eq. (10), you have three vectors,  $\nabla f_{Q}^{(t)}$, $\hat{Q}^\prime$, $M^\prime$, how are they multiplied together? The manner in which they are multiplied together is unclear. Furthermore, the residual term in Eq. (10) should not be a scalar $O(\eta^2)$, but rather a vector. I believe that the second-order error term also depends on the *model size*, i.e., the dimension of $w$.

5. The experiments is only conducted for one instance of $\hat{Q}$ (HTGE Pei et al.), which is insufficient. Additionally, the expression of $\hat{Q}$ from HTGE is missing.

Minor comments:

1. A key reference on the theoretical analysis of STE is missing, specifically: *Yin et al., Understanding Straight-Through Estimator in Training Activation Quantized Neural Networks, ICLR2019.*

**Questions:**

1. In Eq. (5), why the $E^{(t)}$ is defined differently for SGD and Adam?

2. Line 200, "Most multi-bit gradient estimators proposed in the literature are cyclical." Can you specify exactly which estimators are cyclical?

3. Typos in Eq. (9)-(12), $f_{STE}^{(t)}$ should be $\nabla f_{STE}^{(t)}$.

4. In assumption 5.1.1, should $| \hat{Q}^\prime (w) |$ be $\hat{Q}^\prime (w)$ without the absolute value?

5. Does Table 4 suggest that more than 95% quantized weights from $\hat{Q}$-net and STE-net are *identical*? This finding seems counterintuitive.

---

> ### Author Response · Authors · 2024-11-25
> **Mathematical clarifications**
>
> The reviewer enjoyed our presentation and intuitive explanations, but had some concerns about the mathematical details. We are very happy to address all of them below.
>
> ## Weaknesses
>
> 1. **Mathematical Rigor for Weight Movement Differences**
>    We agree with the reviewer on the importance of mathematical rigor in our work. This is why we formalized the concept of weight movement differences with the definition of $E^{(t)}$. Our theorems demonstrate that $E^{(t)}$ is small in the low learning rate regime, which mathematically substantiates the claim that the weight movements are approximately equivalent under these conditions.
>
> 2. **Definition of 'Approximately Equivalent Weight Movement'**
>    The reviewer is correct in their observation that our main metrics, Normalized Weight Alignment Error, and Quantized Weight Agreement were created to measure similarity in weight movements. Both are defined in Section 6.2. Furthermore, the total difference in weight movement is naturally bounded by the sum of $E^{(t)}$ across all steps. Normalized Weight Alignment Error is an aggregate measure of the realized weight movement difference across all steps.
>
> 3. **Error Bounds in Equations (6) and (7)**
>    The error terms are explicitly derived as $O(\eta)$ terms scaled by the gradient differences (which start at zero and remain small as long as the weights remain similar) and $O(\eta^2)$ terms. In typical deep learning applications, the learning rate $\eta$ is very small. Our analysis leverages this fact to show that these error terms remain negligible in practical scenarios, ensuring that they do not accumulate to a level that undermines the main conclusions. If the reviewer is interested in exact calculations of the constants in the theorem statements in some practical scenarios, they can see Appendix H. They are not nearly large enough to overpower the effect of the terms that diminish with $\eta$.
>
> 4. **Mathematical Notation and Derivation Clarifications**
>    - **Scalar Weight Application**: The equations (5)–(12) apply to individual scalar weights, making the use of absolute value notation appropriate.
>    - **Elementwise Quantization**: It is standard practice that quantization functions are applied elementwise to weight tensors, as stated in the first paragraph of Section 2. Note that the theorems apply apply to networks of any size. See the paragraph in section 5.3 titled "The claim applies to networks of any size" for further clarification.
>    - **Residual Term in Equation (10)**: All of these terms are scalars. See the above comments.
>
> 5. **Definition of HGTE**
>    The definition of HGTE (Pei et al., 2023) has now been added to Appendix B for clarity. This should make it easier for readers to understand the experimental results and their connection to the theoretical analysis.
>
> 6. **Missing Reference (Yin et al., 2019)**
>    Thank you for highlighting this reference. While Yin et al. (2019) provides valuable insights into the fundamentals of the STE, it does not address the relationship between custom gradient estimators and the STE, which is the focus of our work. Nevertheless, we have included a citation in Section 2 to acknowledge this important contribution.
>
> ---
>
> ## Questions
>
> 1. **Different Definitions of $E^{(t)}$ in Equation (5)**
>    Adaptive gradient estimators like Adam have a very different behavior from SGD when it comes to gradient estimators. For SGD, the weights are effectively "warped" by the gradient estimator, whereas for Adam, the weights remain approximately the same without any warping. The definitions of $E^{(t)}$ reflect this.
>
> 2. **Cyclical Gradient Estimators**
>    Thank you for pointing this out. Every gradient estimator we mentioned in the paper is cyclical. They are all described in Appendix B.
>
> 3. **Typos in Equations (9)–(12)**
>    You are correct, and we have corrected these notational inconsistencies in the revised manuscript.
>
> 4. **Assumption 5.1.1 and Absolute Value**
>    This is correct! Thank you for fixing this mistake. This does not change the result or the proof.
>
> 5. **Table 4 and Weight Agreement**
>    The quantized weights are identical (there are a small number of choices for quantized weights).
> The full-precision weights are not necessarily identical.

---

> > ### Comment · Reviewer_oPpR · 2024-11-26
> >
> > Thank you to the authors for their response. While some of my questions have been addressed, my primary concerns remain unresolved. Notably, the analysis of the aggregated error over iterations is still lacking. Therefore, I will maintain my current score.

---

> > > ### Author Response · Authors · 2024-11-27
> > >
> > > Thank you for this request, as it has allowed us to strengthen our justification for our experiments. We have devoted a new paragraph at the end of Section 5.3 to aggregate analysis over all iterations. Please see the updated pdf.

---

### Official Review · Reviewer_6p1R · 2024-11-04

**Soundness:** 2
**Presentation:** 2
**Contribution:** 3
**Rating:** 5
**Confidence:** 3

**Summary:**

The authors theoretically analyze the weight difference in QAT when trained with different gradient estimators. Under certain conditions, the authors show that the weight difference is small which means that there's no need to try another gradient estimator other than STE. Empirical results show that the weight difference is small when adopting the proposed weight initialization.

**Strengths:**

1. The claim is strong that other gradient estimators works similar as STE in QAT.
2. Experiments show that the weight difference is small to support the claim.

**Weaknesses:**

1. The mirror room story does not appear closely connected to the theoretical analysis.
2. Assumption 5.1.1 violates Figure 1 where the gradient could be zero.
3. From Table 4, Adam leads to larger weight difference.
4. For more complicated task like ImageNet, the weight difference is much larger than MNIST.

**Questions:**

1. Could you provide unadjusted(A) in Table 4?
2. Besides average error, could you draw a histogram which can better validate the claims?
3. How do you empirically decide when the weight difference is going to affect the conclusion? E.g., Adam shows larger weight difference, while ImageNet shows larger weight difference. It does not seem clear to me they all supports the claim that STE works the same as other gradient estimators on various settings.

---

> ### Author Response · Authors · 2024-11-25
> **Discrepancies in weight divergence across models and optimizers are explained be the Theorem statements**
>
> We thank the reviewer for their thoughtful response, and their recognition that the claims of the
> paper are both strong and well-supported by emprical evidence.
>
> ## Weaknesses
>
> ### Mirror Room Analogy
> We appreciate your feedback on the Mirror Room analogy. The analogy highlights how the movements of the weights in STE-net and $\hat{Q}$-net mirror each other in the case where $E^{(t)}$ is negligible at each step. Our theoretical results demonstrate that $E^{(t)}$ remains small in the low learning rate regime, making the analogy a close reflection of actual behavior. We have updated Section 4 to better describe the connection between the analogy and the theory.
>
> ### Assumption 5.1.1 vs. Figure 1
> Thank you for pointing out this concern. As described in Section 5.3, paragraph 2, and elaborated in Appendix G, our theorems address scenarios where the gradient estimator is zero outside the representable range. In such cases, the behavior of custom gradient estimators can be effectively mimicked by a piecewise-linear estimator, ensuring consistency with our theoretical framework.
>
> ### Weight Difference with Adam
> You are correct that the weight difference is slightly larger when using Adam. SGD is handled by Theorem 5.1 and Adam is handled by Theorem 5.2- since the error terms are different here, we do not expect exactly the same results for both optimizers. Importantly, the difference for Adam remains within a reasonable range relative to the "lr-tweak" baseline.
>
> ### Larger Weight Differences on ImageNet
> The larger weight difference observed in the ImageNet task aligns with our theoretical results. Specifically, the gradient error bound is influenced by the magnitude of the gradient values, which tend to be larger for complex tasks like ImageNet. This relationship is an expected consequence of our theoretical findings and does not detract from the overall validity of the claim.
>
> ---
>
> ## Questions
>
> ### Unadjusted Weights in Table 4
> For adaptive gradient estimators like Adam, no weight initialization adjustment is needed (from the main contributions: "the same result
> holds without any need for adjustment to the learning rate and weight initialization") Therefore, there is no "unadjusted" case to report.
>
> ### Empirical Threshold for Weight Similarity
> Thank you for raising this important point. Unfortunately, there is no universally agreed-upon threshold to determine when weight movements are "approximately the same." To address this ambiguity, we provide metrics for weight similarity and benchmark them against an interpretable baseline (lr-tweak). These comparisons are intended to offer clarity and facilitate interpretation in the absence of a strict boolean criterion. The magnitude of the small differences between the STE and custom gradient estimator does vary depending on the scenario, however the differences result in very little difference in overall performance (table 4). Furthermore, we expect that different optimizers and models will leading to different levels of weight divergence, given the dependence of the error terms in equations (6) and (7) on many different parameters.

---

### Meta-Review · Area_Chair_8tNr · 2024-12-19

**Metareview:**

The authors analyze the impact of gradient estimators on quantization-aware training, concluding that with small learning rates and minor adjustments, most estimators result in similar weight updates. They also claim that the Straight-Through Estimator performs comparably to more complex methods, supporting this claim with theoretical and empirical evidence.

The main issue of this paper is that some main claims are not strongly supported by its theoretical results and its numerical experiments. For example, the authors claim that the weight movement differences will be small if the learning rate is small, so $STE$-net will behave similarly to $\hat Q$-net in SGD and Adam. However, as one of the reviewer pointed out, the error $E^t$ keep increasing as $t$ increases, no matter how small the learning rate is. For a large number of iteration, $E^t$ can still be very large. The review requested some analysis on the cumulative error. However, the authors just provide the formula of the cumulative error on page 7 which is basically $O(t\eta)$ and is not necessarily small for a large $t$. Moreover, $\eta$ is not necessarily always small when training a model. It may decrease with $t$ so it can be large for the first few iterations.

Also, the authors claimed that practitioners only need to use the Straight-Through Estimator because other estimators are similar to STE after modifying the learning rate and initialization. However, one reviewer pointed out that this claim is not supported by the current experiments. It needs a direct comparison with ""custom gradient estimators in the settings where they were originally proposed"".

I also think this paper is not well written with many abuses of math notations and missing definitions. The review team has pointed out some but there are still many others. For example, what is the definition of "$round()$" function? Also, $STE$-net and $\hat Q$-net are defined in words rather than using clear math formula.

**Additional Comments On Reviewer Discussion:**

All reviewers provide useful feedbacks and their reviews are evenly weighted.

I especially appreciate Reviewer oPpR and Reviewer KqAp who pointed out the critical issues on this paper I mentioned in the meta review. In particular, Reviewer oPpR found that the theory in this paper does support the claim, e.g., because the error can still grows as the number of iterations increases. Reviewer KqAp pointed out that the current experiments are not sufficient to convince practitioners to only use STE estimator.

---

### Decision · Program_Chairs · 2025-01-22

Reject